



# Droughts in the area of Poland in recent centuries

Rajmund Przybylak[1] ORCID: 0000-0003-4101-6116, Piotr Oliński[2] ORCID: 0000-0003-1428-0800, Marcin Koprowski[3] ORCID: 0000-0002-0583-4165, Janusz Filipiak[4] ORCID: 0000-0002-4491-3886, Aleksandra Pospieszyńska[1] ORCID: 0000-0003-2532-7168, Waldemar Chorążyczewski[2] ORCID: 0000-0002-0063-0032, Radosław Puchałka[3] ORCID: 0000-0002-4764-0705, and Henryk P. Dąbrowski[5] ORCID:0000-0002-8846-5042

[1] Department of Meteorology and Climatology, Faculty of Earth Sciences, Nicolaus Copernicus University, Toruń, Poland

[2] Department of Medieval History, Institute of History and Archival Sciences, Faculty of History, Nicolaus Copernicus University in Toruń, Poland

[3] Department of Ecology and Biogeography, Faculty of Biology and Environmental Protection, Nicolaus Copernicus University, Toruń, Poland

[4] Department of Meteorology and Climatology, Institute of Geography, Faculty of Oceanography and Geography, University of Gdansk, Poland

[5] Dendroarchaeological Laboratory, Archaeological Museum in Biskupin, Biskupin, Poland

*Correspondence to:* R. Przybylak (rp11@umk.pl)

**Abstract**: The paper presents the main features of droughts in Poland in recent centuries, including their frequency of occurrence, coverage, duration and intensity. For this purpose both proxy data (documentary and dendrochronological) and instrumental measurements of precipitation were used. The reconstructions of droughts based on all the mentioned sources of data covered the period 996–2015. Examples of megadroughts were also chosen using documentary evidence, and some of them were described.

Various documentary sources have been used to identify droughts in the area of Poland in period 1451–1800 and to estimate their intensity, spatial coverage and duration. Twenty-two local chronologies of trees (pine, oak, and fir) from Poland were taken into account for detecting negative pointer years (exceptionally narrow rings). The longest chronology covers the years 996–1986 and was constructed for eastern Pomerania. The delimitation of droughts based on instrumental data (eight long-term precipitation series) was conducted using two independent approaches. In the first approach we used the globally and nationally popular Standard Precipitation Index (SPI), which was calculated for 1-, 3-, and 24-month time scales. Thus, three categories of droughts were analysed: meteorological (SPI1), agricultural (SPI3) and hydrological (SPI24). For delimitation of droughts (dry months), the criteria used were those proposed by McKee (1993) and modified for the climate conditions of Poland by Łabędzki (2007). Droughts were divided into three categories based on the following SPI values: moderate droughts (-0.50 to -1.49), severe (-1.50 to -1.99), and extreme (≤-2.00). The second approach includes the new proposed method for distinguishing droughts and quantitatively estimating their intensity and duration.

More than one hundred droughts were found in documentary sources from the mid-15th century to the end of the 18th century, including 17 megadroughts. A greater-than-average number of droughts was observed in the second halves of the 17th century, and of the 18th century in particular. Dendrochronological data confirmed this general tendency in the mentioned period. The clearly greatest number of negative pointer years occurred in the 18th century and then in the period 1451–1500. In the period 996–2015, a total of 758 negative pointer years were recorded.

Analysis of SPI (including its lowest values, i.e. droughts) showed that the long-term frequency of droughts in Poland has been stable in the last two or three centuries. Extreme and severe droughts were most frequent in the coastal part of Poland and in Silesia. Most droughts had a duration of two months (about 60–70%), or 3–4 months (10–20%). Frequencies of droughts with a duration of 5 and more months were lower than 10%. The longest droughts had a duration of 7–8 months. The frequency of droughts of all categories in Poland in the period 1722–2015 was greatest in winter. This fact should be taken into account when analysing droughts delimited using documentary evidence. In Poland in 1451–1800, in light of this sort of information, droughts in spring and summer clearly dominated, while only three winter droughts were mentioned.

The occurrence of negative pointer years (a good proxy for droughts) was compared with droughts delimited based on documentary and instrumental data. A good correspondence was found between the timing of occurrence of droughts identified using all three kinds of data (sources).



## 1 Introduction

The increase in degree of global warming that has been observed in recent decades also influences characteristic changes in the occurrence and intensity of precipitation (IPCC, 2013). Although precipitation totals are slightly greater from year to year in some regions, frequency of precipitation is getting lower, while its intensity is increasing. As a result, breaks between precipitation episodes are getting longer and longer, which significantly favours the occurrence of droughts. The majority of statistical analyses conducted for the entire world (Dai and Trenberth, 1998; Dai et al., 2004; Dai, 2011a, b, 2013; IPCC, 2013) and its different regions (see, e.g., Held et al., 2005; Alexander et al., 2006; Bartholy and Pongracz, 2007; Łabędzki, 2007; Brázdil et al., 2009; Seneviratne et al., 2012; NAS, 2013; Miles et al., 2015; Osuch et al., 2016; Bąk and Kubiak-Wójcicka, 2017) usually confirm the rising frequency and intensity of droughts in recent decades. However, some authors document that this change for the entire globe is not as big as is presented in the abovementioned publications (Sheffield et al., 2012). They argue that overestimation of the rate of change of global droughts is related to the shortcomings (simplifications) of the Palmer Drought Severity Index (PDSI) used for this purpose. They write: "The simplicity of the PDSI, which is calculated from a simple water-balance model forced by monthly precipitation and temperature data, makes it an attractive tool in large-scale drought assessments, but may give biased results in the context of climate change." Nevertheless, a greater or lesser increase in frequency of droughts in global scale has been observed in recent decades. Moreover, climatic models project that this tendency will also be seen in the entire 21$^{st}$ century. It is very likely that droughts will be not only more frequent, but also more intense in many regions, but particularly in areas with dry conditions in today's climate (IPCC, 2013). For this reason, the study of drought occurrence and its intensity is very important, in particular when its manifold negative socio-economic consequences are taken into account. Many aspects dealing with drought (definition; kinds – meteorological, agricultural, hydrological, socio-economic; quantitative ways of measurement; socio-economic consequences; etc.) were described recently in many publications (e.g. Wilhite and Glantz, 1985; Tate and Gustard, 2000; Herweijer et al., 2007; Mishra and Singh, 2010; Dai 2011a; Brázdil et al., 2013, 2018; IPCC, 2014; Fragoso et al., 2018) and therefore a brief overview is omitted here.

To estimate how unprecedented is the scale of climate drying in recent decades, a longer perspective is needed. Therefore, in recent decades quite a lot of drought reconstructions encompassing almost the entire millennium, or the shorter historical, pre-industrial period, were constructed for different greater or smaller regions (e.g. Inglot, 1968; Piervitali and Colacino, 2001; Cook et al., 2004, 2010, 2015; Herweijer et al., 2007; Pfister et al., 2006; Brewer et al.,



2007; Domínguez-Castro et al., 2008, 2010; Woodhouse et al., 2010; Brázdil et al., 2013, 2016,
2018 (see references herein); Dobrovolný et al., 2015; Fragoso et al., 2018; Hanel et al., 2018).

3         What is the state of knowledge about droughts occurrence and intensity in Poland – the area

that is the object of our studies in the paper? It must be said that for the instrumental period, and
in particular for the period after World War II, the knowledge is good. Papers have been published
analysing: 1) classification of drought types and the development of drought indices (Bąk and
Łabędzki, 2002; Łabędzki, 2007; Łabędzki and Kanecka-Geszke, 2009; Tokarczyk, 2013;
Łabędzki and Bąk, 2014); 2) tendencies in drought occurrence and intensity (Farat et al., 1998;
Magier et al., 2000; Łabędzki, 2007; Kalbarczyk, 2010; Bartczak et al., 2014; Radzka, 2015;
Wypych et al., 2015; Bąk and Kubiak-Wójcicka, 2017); 3) monitoring of drought conditions
(Łabędzki, 2006; Doroszewski et al., 2008, 2012; Tokarczyk and Szalińska, 2013; IMGW, 2014;
ITP, 2014; Łabędzki and Bąk, 2014); and 4) drought hazard assessment for periods when
observations are available (Łabędzki, 2009; Tokarczyk and Szalińska, 2014). In recent years the
influence of future climate change on the occurrence of droughts in Poland in the 21$^{st}$ century has
also been addressed (Liszewska et al., 2012; Osuch et al., 2012, 2016). On the other hand, little is
known about drought occurrence in the pre-instrumental and early instrumental periods in Poland.
Generally, only one team of researchers under the direction of professor Stefan Inglot of Wrocław
University was focusing on this issue, in the 1960s. As a result, a first attempted chronology of
droughts for the 16$^{th}$ to mid-19$^{th}$ century was proposed based on documentary evidence (Inglot,

20   1968).

21        Drought is the one of the most stressful factors for trees (Vitas, 2001; Allen et al., 2010; Sohar

et al., 2013). The measurement of tree ring widths is one of the ways to study the effect of climate
parameters on trees (Zielski et al., 2010). Some factors such as frost or summer drought may have
an immediate effect on ring width, whereas other factors, such as winter drought, may have a
delayed effect on ring widths. This delayed effect occurs because the meristematic tissues are
dormant during the winter months in temperate and cold climates. The effect of different factors
is seen as variations in ring size and structure, which change systematically, or vary slowly
throughout the life of the tree (Fritts, 1976). The effect of drought on tree rings is observed as
narrow rings (Koprowski et al., 2012; Opała, 2015). The relationships are significant enough to
reconstruct drought in Finland (Helama and Lindholm, 2003), Sweden (Seftigen et al., 2013) and
Czech Republic (Dobrovolný et al., 2015). Therefore, we have assumed that information derived
from tree rings can complement the existing knowledge about past droughts in Poland.

33        Although in the last three decades many climate reconstructions for the last millennium have

been conducted for Poland (see Przybylak et al., 2005 or Przybylak, 2016 for a review), droughts
were not analysed. Therefore, to fill this important gap we decided to investigate them in more
detail than was done by the Inglot's team. Moreover, for this purpose we used more sorts of proxy



data (not only documentary but also dendrochronological). The reconstructions of droughts based on all the mentioned sources of data covered the period 996–2015. Thus, the main aim of the present paper is to present the main features of drought occurrence, duration and intensity in the area of Poland in this period. Section 2 describes all the kinds of data used and their quality. Section 3 addresses the methods used in this study, including drought indices. Section 4 presents the results of three reconstructions of droughts derived from 1) documentary, 2) instrumental, and 3) dendrochronological data. Examples of megadroughts are also analysed here. The results obtained are discussed in Section 5, and main conclusions in the last section.

## 2  Data

### 2.1. Documentary data

Records on drought can be found in many different historical sources from Polish territories. Their number has significantly increased since the mid-15[th] century, which is why the mid-15[th] century was adopted as the initial chronological boundary for the reconstruction of the number and intensity of droughts in the Polish territory using documentary evidence. Below we describe the types of historical sources used to reconstruct droughts in Poland.

Records of droughts in the Polish territory are most often found in narrative sources – chronicles, yearbooks, memoirs, diaries, travel accounts. The information included in these sources has a varying degree of accuracy. Often only one account concerning drought appeared, such as, for example, "magna siccitas". In many of the records, however, more detailed descriptions of the course of droughts and accompanying phenomena were given. In the ancient sources droughts were described above all when their manifestations were very clear and when they had an impact on economic and social life. Another group of sources used by us are daily records that have the character of meteorological observations. Sometimes, they were prepared by scholars such as professors of the Jagiellonian University Marcin Biem (ca. 1470–1540) and Michał of Wiślica (1499–1575), who conducted such observations in Kraków from 1499 to 1531 and from 1534 to 1551 (Limanówka, 2001), or townsmen with scientific ambitions such as Gottfried Reyger (1704–1788), who began his observations in Gdańsk in 1721 as a 17-year-old and continued them later, among others as a member of the *Naturforschende Gesellschaft* in Gdańsk until 1786 (Filipiak et al., 2019). Sometimes daily observations were conducted by amateurs, the best example of which are the records of the Polish nobleman from the eastern territories of the Polish–Lithuanian Commonwealth, Jan Antoni Chrapowicki, which were conducted for the years 1656–1685 (Nowosad et al., 2007). Sources of this kind are nonetheless relatively rare.



The correspondence, the manuscript press ("written newspapers") and printed press were
also used in the reconstruction of droughts. In the case of written newspapers, these are often
records similar to those that appear in chronicles. They were drawn up on a regular basis, which
increases their credibility. They provided news from the region, as well as information coming
from other countries, e.g. from Lviv, from which a newswriter in 1698 wrote: "in these countries
shamefully there are great droughts, for which reason we sowed very little for the winter, because
you cannot cut the land with the ploughshare" [w tych krajach chaniebnie [! – emphasis added]
susze wielkie, dla których na zimę bardzo mało siano, bo nie podobna lemieszem ukroić ziemię"]
(Maliszewski, 2018). Other sources that turned out to be useful for the implementation of our
project were official files (e.g. protocols from meetings of the regional dietines and the Sejm,
treasury registers, inspection reports) documenting activities undertaken, e.g. in connection with
droughts and fires. They reported requests for financial support in connection with drought, tax
exemption requests, etc. In economic files one can find explanations for low harvests, which
occurred for example due to drought. There are a few sources concerning religious behaviours in
which, for example, the organisation of prayers asking for rain or describing the end of a drought
were described. When such accounts appeared, it can be assumed that the drought must have been
severe for people and the environment.
In addition to the above mentioned historical sources collected during the queries in Polish,
Lithuanian, Ukrainian and German archives, the authors used several published collections (of
varied quality) of historical sources concerning the climate research in the period from the 10$^{th}$
century to the end of the 18$^{th}$ for Poland, Europe Central or selected regions of Central Europe.
They include: the period from the 10$^{th}$ century to the end of the 16$^{th}$ (Girguś et al., 1965); the
Middle Ages (Malewicz, 1980); 1450–1586 (Walawender, 1935); the years 1648–1696
(Namaczyńska, 1937); and 1772–1848 (Szewczuk, 1939). In the last 20 years, two databases
containing over ten thousand weather records were made available in universities in Toruń and
Wrocław as part of cooperation between climatologists and historians. They have been used many
times to study Poland's climate in historical times (Wójcik et al., 2000; Przybylak et al., 2001,
2004, 2005, 2010; Przybylak, 2011, 2016); they have also contributed to widening the scope of
this research.
To sum up, for the purpose of this research over 200 accounts referring directly to droughts
and prolonged shortages of rainfall were used, along with a few hundred more descriptions from
everyday weather observations, the use and critical elaboration of which allowed periods of
drought to be indicated. The state of the preservation of sources for particular periods and for
individual regions is uneven. Most of them describe droughts in Silesia, Pomerania and Lesser
Poland. A large number of entries refers to droughts affecting the whole territory of Poland. In the
case of Silesia, the distribution of sources is fairly even for the whole period; in the case of other

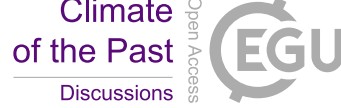

regions their number increases with successive ages. The only exception is the first half of the 17[th]
century, in which the number of preserved records is definitely smaller. To some extent, this was
affected by the losses in the state of preserved sources that occurred during the Swedish invasion
on Polish territories in 1655-1660. Many sources from the first half of the 17[th] century were then
destroyed as a result of military actions.
The accuracy scale of the collected information is variable. Some accounts provide quite
precise information concerning the duration of the drought, even to the accuracy of one day, while
others are definitely more general – they only indicate the existence of a drought in a given year.
It very often occurs that one drought is described in several, or sometimes even several dozen,
independent sources, which confirms its high intensity.
To assess the credibility of individual records, it was necessary to conduct a critical source
analysis, in which it turned out that sometimes even short accounts provided very important and
reliable information, while other records with a similar structure proved to be wrong due to the
fact that, e.g., the year of the occurrence of the drought was changed (e.g. by one year) when the
information was being copied from another, earlier source. The sources containing daily records,
as in the case of the memoirs of A. Chrapowicki or G. Reyger required a different treatment. It
was possible to count the days with precipitation and without precipitation along with a very
precise indication of the duration of the droughts.
**2.2.    Dendrochronological data**
We used 22 chronologies (17 oak chronologies, 5 pine chronologies and 1 fir chronology) from
different locations in Poland to detect pointer years (Table 1, Fig. 1). Table 1 presents a list of
them, including also time coverage and sources. As results from this Table, the longest
chronology available to us covers the years 996–1986 and was constructed for eastern Pomerania
(Site 5). For Upper Silesia (Sites 16 and 18) and Lesser Poland (Sites 21 and 22), the pointer
years were detected by Opała and Mendecki (2014) and Opała (2015) for Upper Silesia, and by
Szychowska-Krąpiec (2010) for Lesser Poland (Table 1, Fig. 1).
Table 1. Basic characteristic of the chronologies used for pointer year analysis. Location
of natural-forest regions (Zielony and Kliczkowska, 2010) and sites is shown in Fig. 1

| Site number | Site name | Time span | Species | Source |
|---|---|---|---|---|
| Region I (Baltic Province) | | | | |
| Site 1 | Koszalin | 1782–1987 | Oak | https://www.ncdc.noaa.gov/ (Ważny, 1990) |
| Site 2 | Gdańsk | 1762–1986 | Oak | https://www.ncdc.noaa.gov/ (Ważny, 1990) |
| Site 3 | Wolin | 1554–1987 | Oak | https://www.ncdc.noaa.gov/ (Ważny, 1990) |





| Site 4 | Gdańsk | 1175–1396 | Oak | Dąbrowski HP, unpublished |
|---|---|---|---|---|
| Site 5 | western Pomerania | 996–1986 | Oak | https://www.ncdc.noaa.gov/ (Ważny, 1990) |
| Region II (Masuria-Podlasie Province) | | | | |
| Site 6 | Gołdap | 1871–1987 | Oak | https://www.ncdc.noaa.gov/ (Ważny, 1990) |
| Site 7 | Suwałki | 1861–1987 | Oak | https://www.ncdc.noaa.gov/ (Ważny, 1990) |
| Site 8 | Hajnówka | 1720–1985 | Oak | https://www.ncdc.noaa.gov/ (Ważny, 1990) |
| Region III (Greater Poland-Pomerania Province) | | | | |
| Site 9 | Poznań | 1836–1987 | Oak | https://www.ncdc.noaa.gov/ (Ważny, 1990) |
| Site 10 | Zielona Góra | 1774–1987 | Oak | https://www.ncdc.noaa.gov/ (Ważny, 1990) |
| Site 11 | Toruń | 1714–2011 | Oak | Puchałka et al., 2016 |
| Site 12 | Tuchola | 1249–1490 | Pine | Dąbrowski HP, unpublished |
| Site 13 | Kuyavia-Pomerania | 1169–2015 | Pine | Koprowski et al., 2012 |
| Site 14 | Chojnice | 1100–1468 | Oak | Dąbrowski HP, unpublished |
| Region IV (Masovia-Podlasie Province) | | | | |
| Site 15 | Warszawa | 1690–1985 | Oak | https://www.ncdc.noaa.gov/ (Ważny, 1990) |
| Region V (Silesia Province) | | | | |
| Site 16 | Upper Silesia | 1770–2010 | Pine and oak | Opała and Mendecki, 2014 |
| Site 17 | Wrocław | 1727–1987 | Oak | https://www.ncdc.noaa.gov/ (Ważny, 1990) |
| Site 18 | Upper Silesia | 1568–2010 | Pine | Opała, 2015 |
| Region VI (Lesser Poland Province) | | | | |
| Site 19 | Kraków | 1792–1986 | Oak | https://www.ncdc.noaa.gov/ (Ważny, 1990) |
| Site 20 | Kosobudy | 1782–1989 | Oak | https://www.ncdc.noaa.gov/ (Ważny, 1990) |
| Site 21 | Lesser Poland | 1109–2004 | Pine | Szychowska-Krąpiec, 2010 |
| Site 22 | Lesser Poland | 1109–2006 | Fir | Szychowska-Krąpiec, 2010 |



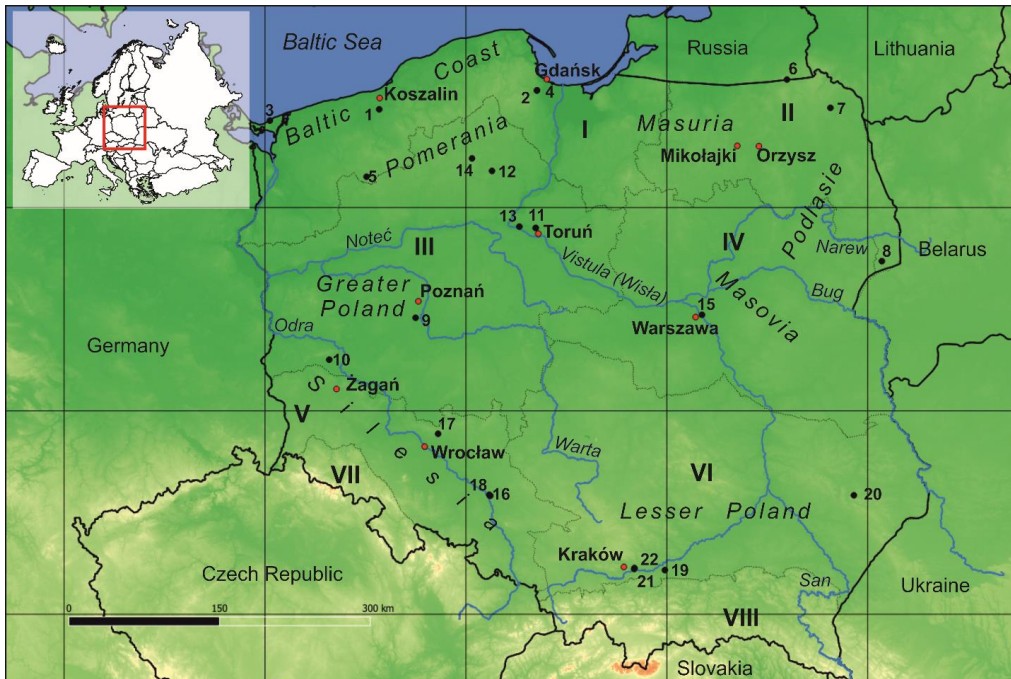

Fig. 1. Location of dendrochronological sites (black dots, for more details see Table 1) and
meteorological series (red dots, for more details see Table 2) used in the study
**2.3. Instrumental data**
**2.3.1. Isolated series**
The number of known precipitation series and whose beginnings date back to earlier than the 20th
century is very limited. There are only a dozen of those begun before 1800. Efforts to organise
meteorological measurements in Poland were made relatively early in comparison to other
European countries. The country's complicated history (e.g. many wars and changes of borders)
has resulted in the loss of the majority of sources collected in the archives, in many cases
irretrievably. However, actions to restore the long measurement series based on the discovered
collections have been taken for a few selected locations.
The oldest surviving results of instrumental precipitation series in Poland come from
Gdańsk and are dated to the first half of the 18th century. In January 1739, Michael Christoph
Hanov, a mathematician and physician, started daily observations of weather phenomena and
measurements of a dozen meteorological elements, including precipitation. The results of his
efforts were published in the newspaper *Danziger Erfahrungen* on a weekly basis. Hanov
presented the complete series in his manuscript *Wetter Beobachtungen in Danzig 1739–1773*.
Hanov's instrumental series was accompanied by the notes from a weather chronicle
authored by Gottfried Reyger. He started systematic observations of the weather in Gdańsk in





December 1721 and carried them out until the mid-1786. The results of observations were used mainly to study how climate affects the development of plants. Reyger published the outcomes of his studies in *Die Beschaffenheit der Witterung in Danzig vom Jahr 1722 bis 1769 beobachtet nach ihren Veränderungen und Ursachen erwogen* (Reyger, 1770) and in *Die Beschaffenheit der Witterung in Danzig. Zweyter Theil vom Jahr 1770 bis 1786, nebst Zustätzen zur Danziger Flora* (Reyger, 1788).

Reyger usually presented remarks on general weather conditions supplemented by some additional data. Months were usually described in a qualitative, even aggregate, manner. His notes were very detailed and even the weather of the particular days or weeks was very often characterised. Reyger paid special attention to particularly important weather and climate phenomena (heavy rain, floods, droughts, and heat and cold waves). His notes after 1783 (Hanov's death) were more accurate. Despite the lack of measured values of precipitation, detailed data on the monthly number of rainfall and snowfall were presented (for more details including the reconstruction of the air temperature and precipitation series since 1721 see Filipiak et al. [2019]). Some sources suggest an even earlier date for the beginning of Reyger's instrumental observations (Hellmann, 1883, after Rojecki, 1965). Besides the short description in the mentioned literature no other proof of such activity is available.

### 2.3.2. Long-term continuous series

The series from Wrocław (formerly Breslau) that commenced in 1791 (Bryś and Bryś, 2010) is the longest continuous Polish precipitation series. For the purpose of the present paper we prolonged this series until 1781 based on precipitation measurements in Żagań (formerly Sagan) within the Mannheim network of stations established for Europe and North America by the Palatine Meteorological Society in 1780 (Przybylak et al., 2014). The cited authors proved that there exist high correlations between the precipitation series from both places. Source data from Żagań were taken from the publication *Ephemerides Societatis Meteorologicae Palatinae, 1783–1795*. In addition, we must say that the Wrocław series is the only continuous series to have begun before 1800 in the area currently belonging to Poland. The best known long-term climatological series in Poland is the one from Kraków that commenced in 1792. The work on completing the collections of the Kraków series continue till the present day, the effect of which are reconstructions of monthly values of precipitation sums since 1863 (Twardosz, 2005, 2007). As for other Polish cities, Lorenc (2000) performed a homogenisation of series of monthly precipitation totals of Warszawa (Warsaw) since 1813. Miętus (2002) reconstructed atmospheric precipitation sums from Koszalin (formerly Köslin) since 1848. In another paper, Kożuchowski and Miętus (1996) presented series of precipitation totals in Szczecin (formerly Stettin) since 1848.

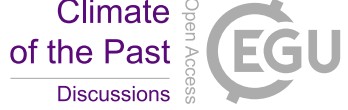

In 2011 a reconstruction was performed of the precipitation series from Gdańsk in 1880–2008
(further extended to 1851) (Filipiak, 2011). During the CLIMPOL project (Climate of northern
Poland during the last 1000 years: Constraining the future with the past) Filipiak reconstructed the
series of monthly precipitation totals since 1891 for Lake Żabińskie in NE Poland (54°07' N;
21°59' E) (Larocque-Tobler et al., 2015). Further, the series of Orzysz (formerly Arys) and
Mikołajki (formerly Nikolaiken), also in NE Poland, were collected for the years 1830–1904 and
since 1889, respectively. As both stations are located very close to one another (approximately 20
km) these two series have very much in common. The correlation coefficient calculated for the
overlapping periods 1889–1904 and 1981–2015 exceeds 0.85. Thus we decided to combine both
series: data from Orzysz covers the period between 1830 and 1890, the later data comes from
Mikołajki. A couple of series, e.g. Poznań (formerly Posen), Toruń (formerly Thorn), Racibórz
(formerly Ratibor), Śnieżka (formerly Schneekoppe), began around the middle of the 19[th] century
and are available in yearbooks that were initially released by the Royal Prussian Meteorological
Institute (Königlich Preussischen Meteorologischen Institut), then since 1918 by the Polish
National Meteorological and Hydrological Service (PIM until 1945, further PIHM and finally,
after 1972 IMGW). The complete list of instrumental series employed in the current research and
their sources are presented in Table 2.
Table 2. List of sites, their locations and periods covered by series of monthly precipitation totals
used in the paper

| No. | Station | Geographical region | Observation period | Location ($\varphi$, $\lambda$, h) | Sources of data |
|---|---|---|---|---|---|
| Isolated series | | | | | |
| 1a | Gdańsk [*] | 1 | 1722–1786 | 54°20'N 18°40'E 13 m a.s.l. | Reyger (1770, 1788) and Filipiak et al. (2019) for the periods 1722–1738 and 1773–1786; Hanov (1773) for the period 1739–1773 |
| Long-term continuous series | | | | | |
| 1b | Gdańsk | 1 | 1851–2015 | 54°20'N 18°40'E 13 m a.s.l. | Filipiak (2010 modified 2018) for the whole period |
| 2 | Koszalin | 1 | 1851–2015 | 54°12'N 16°11'E 46 m a.s.l. | Reichsamt für Wetterdienst (1939) for the period 1851–1930 corrected by Miętus (2002); Miętus (2002) for the period 1931–1990; Central Database of Historical Data of IMGW-PIB (Polish National Meteorological and Hydrological Service) for years 1991–2015 |
| 3a | Orzysz | 2 | 1830–1890 | 53°48'N 21°56'E 122 m a.s.l. | Dove (1851) for the period 1830–1850; Reichsamt für Wetterdienst (1939) for the years 1851–1904 |





| 3b | Mikołajki | 2 | 1891–2015 | 53°48'N 21°34'E 116 m a.s.l. | Central Database of Historical Data of IMGW-PIB for the whole period |
|----|-----------|---|-----------|------------------------------|----------------------------------------------------------------------|
| 4 | Toruń | 3 | 1871–2015 | 53°01'N 18°36'E 60 m a.s.l. | Pospieszyńska and Przybylak (2013) for the period 1871–2010; Central Database of Historical Data of IMGW-PIB for years 2011–2015 |
| 5 | Poznań | 3 | 1848–2015 | 52°25'N 16°56'E 66 m a.s.l. | Dove (1851) for the period 1848–1850; Central Database of Historical Data of IMGW-PIB for the years 1851–2015 |
| 6 | Warszawa | 4 | 1813–2015 | 52°13'N 21°01'E 97 m a.s.l. | Lorenc (2000, 2007) for the years 1813–1999; Central Database of Historical Data of IMGW-PIB for the years 2000–2015 |
| 7a | Żagań | 5 | 1781–1790 | 51°37'N 15°19'E 102 m a.s.l. | *Ephemerides Societatis Meteorologicae Palatinae, 1783–1795* for the whole period |
| 7b | Wrocław | 5 | 1791–2015 | 51°07'N 17°05'E 120 m a.s.l. | Bryś and Bryś (2010) for the years 1791–2000; Central Database of Historical Data of IMGW-PIB for the years 2001–2015 |
| 8 | Kraków | 6 | 1876–2015 | 50°04'N 19°58'E 216 m a.s.l. | Kożuchowski (1985) for the period 1876–1900, Twardosz (2007) for the years 1901–2000, Central Database of Historical Data of IMGW-PIB for the years 2001–2015 |

Key: geographical regions: 1 – Baltic Coast – Pomerania, 2 – Masuria – Podlasie, 3 – Greater Poland, 4 – Masovia, 5 – Silesia, 6 – Lesser Poland

[*]the series for periods 1722–1738 and 1773–1786 were reconstructed based on Reyger's weather chronicle

## 3. Methods

### 3.1. Documentary data

The collected historical sources informing about droughts were evaluated according to a three-level scale, taking into account, first of all, signalled manifestations and consequences of the drought and its duration. The droughts were divided into "extreme", "severe" and "moderate".

Extreme droughts (-3) constituted periods of no rainfall or very scarce rainfall that were long-lasting – they lasted at least one season (2–3 months and more). The principle was adopted that extreme droughts should be recorded in sources from two regions or more; even in view of the absence of sources this allows us to assume that these were droughts of an exceptional nature, having been noted by many writers. Such an extreme drought of 1473 was described, among





others, in the "Annals" by Jan Długosz and, for example, in the local chronicle of Wrocław by
Nicolaus Pol. When the source information indicated an extreme drought, but at the same time
there appeared information, for example, about the elevated state of water or floods, which may
have indicated heavier precipitation especially in the summer season, it was concluded that no
extreme drought had taken place. In agricultural terms, extreme droughts contributed to much
earlier cereal harvests; they often seriously threatened the growth and size of yield, as was
mentioned in the sources. Descriptions of extreme droughts usually contain several permanent
elements: severe water shortages, fires, the destruction of crops; sometimes there also appeared
records about the fact that people did not remember a similar drought having occurred in their
lifetime. These droughts caused water reservoirs – ponds and lakes – to dry up completely.
Sometimes, and probably in an exaggerated way, sources reported the drying up of smaller rivers.
During extreme droughts, there were frequent records of persistent very low water levels
in the largest rivers – the Odra and the Vistula (Table 3). The result was a lack of water for people
and animals, halting the work of water mills in whole provinces. The consequences of drought
were underlined – particularly a lack of food and high prices. Numerous fires broke out in cities,
villages and forests. The sources used such phrases as "*estas ferventissima et siccitas inaudita*"
[very hot and incredible summer drought], "*sidere solari plus solito effervescente et nullas dante*
*pluvias*" [extraordinary sun heat and continuous drought], "*unaufhörlich trockene Witterung*"
[unbelievably dry weather], "*alle Bäche vertrockneten*" [all streams dried up] and the like,
underlining the extreme nature of the drought. Superlative adjectives were very often used.
Table 3. Examples of descriptions of extreme droughts in 15th–17th-century sources

| Year | Description | Translation | Source |
|---|---|---|---|
| 1463 | [...] *fuit magnus calor et arditas, ita quod sylvae, nemora et montana incenderentur, ex voragine ignis pro magna parte absumptae* | [...] there were such great heat and drought that forests, groves and mountain vegetation burned, and were largely destroyed by the fire | Rocznik wrocławski dawny, MPH, vol. 3, p. 686 |
| 1473 | [...] *caumata et penuriam aquarum, adeo ut perennes aquae verterentur in aridam, et flumina Poloniae principalia ubique fuerunt permeabilia, insignis.* [...] *Fumabant in universis Poloniae regionibus* | [...] hot weather and a lack of water, to such an extent that the places where there had always been water dried up everywhere, and the main Polish rivers could be crossed everywhere. [...] Forests, | Długosz, vol. 12, p. 336 |



| | silvae, borrae, arbusta, saltus, irremediabili igne, nec ante rescindi flamma poterat, donec ignis etiam radicem arborum voraret, ex quo ubique fragor ruentium saltuum audiebatur. Apum quoque et alveariorum arbores plurimae deletae, segetes vernales exterminatae siccitate. | woods, thickets and forested hills burnt with fire; there was no way to put it out, and it was impossible to extinguish the flame before the fire even devoured the root of the trees; from here you could hear the clatter of collapsing thickets. Very numerous bee and beekeeping trees were destroyed, and many spring crops were destroyed due to drought. | |
|---|---|---|---|
| 1540 | [...] *fuit in aestate horrenda siccitas adeo, ut silices, montes et valles quasi igne flagrarent, duravit haec siccitas usque ad hyemem.* | [...] in the summer there was such a terrible drought that the rocks, mountains and valleys were burned down with fire; this drought lasted until winter | Archivum vetus et novum ecclesiae archipresbyteralis Heilsbergensis, in: MHW, vol. 8, p. 597 |
| 1561 | *Im Julio und Augusto war es sehr dürre und dürre Winde, dass das Wasser sehr austrocknete. Die Oder war klein, dass es keinem Mann gedachte. Viel Brunnen trockneten aus.* | In July and August there were dry and very dry winds, so that the water completely dried out. The Odra became shallow as it had never been before. Many wells dried up. | Pol, vol. 4, p. 17 |
| 1575 | *At in Polonia inaudita fere siccitas vere, aestate, autumno et hyeme denique aestivalium segetum, quas arefecerat, penuriam fecit, amnium vero undas adeo minuit, ut iis passim fere privaretur ipsaque Vistula infra Dobrinum multis locis vadabilis fieret, unde nec sal e* | However, in Poland, a truly unbelievable drought, in summer, autumn and winter, along with spring crops that had dried up, caused poverty[;] the level of the water in rivers had fallen so much that everywhere the rivers almost disappeared, while the Vistula | Orzelski, in: SRP, vol. 22, p. 360 |



| | | | |
|---|---|---|---|
| | *Russia per Sanum in Vistulam permeari potuit.* | in many areas below Dobrzyń became quite shallow, and it was not possible to transport any salt from Ruthenia through the San to the Vistula. | |
| 1590 | *Ist ein sehr heisser truckener Sommer gewesen, also, dass auch die Landflüsse, als der Bober, Queiss, Katzbach, Weida, Olau, Lohe, und andere mehr gänzlich ausgetrucknet. Die Oder ist auch so klein worden, dass man sie an allen Orten durchwatten können.* | The summer was so hot [and] dry that national rivers like the Bóbr, the Kwisa, the Kaczawa, the Widawa, the Oława, the Ślęża [Silesia, auth. suppl.] and many others dried up completely. The Odra also became very shallow, so you could cross it anywhere. | Pol, vol. 4, p. 156 |
| | *38 Wochen regnete es nicht. Die Flüsse trockneten aus.* | It did not rain for 38 weeks. The rivers dried up. | Reinhold, 1846, p. 143 |
| | *Zacken und andere Flüsse trockneten völlig aus* | The Kamienna and other rivers dried up completely. | Bergemann, J.G., 1830a, p. 84 |
| | *Der Bober* [river] *trocknete infolge starker Hitze ganz aus.* | The Bóbr [river] dried up completely due to severe drought. | Bergemann, J.G. 1830b, vol. 3, p. 85 |
| 1653 | *In Monath Maii fiel ein dürres Wetter ein, und dauerte biss Ende August. Die alle Bäche vertrockneten, auch Flachs und Gerste verdorrete.* | In the month of May the dry weather began and lasted until the end of August. All streams dried up, as did flax and barley. | Gomolcke, p. 53 |
| 1676 | *Tego roku straszne Panowały Susze, że zboża wypalało w polach.* | That year a terrible drought took place so that crops burnt in the fields. | Muz. Nar. w Krakowie rps. MNKr. 169, p. 82 |
| 1683 | *Im Jahre 1683 entstand durch die grosse Dürre und den Misswachs eine starke Theuerung und ein fast* | In 1683, due to the great drought and poor growth [of grain], high prices and almost | Pisański, Beschreibung der Stadt |



| Year | Description | Translation | Source |
|---|---|---|---|
| | *gäntzlicher Mangel an Getreyde.* | complete lack of grain prevailed. | Johannisburg, p. 96 |
| 1684 | *[...] folgete auf Johanni [24.06] eine grosse anhaltende Hitze darauf; davon das Erdreich dermassen dürre wurde, dass das Sommer-Getreyde, Flachs, und Grass, gantz zurücke geblieben, das Winter-Korn an vielen Orten überreiffte, ehe es sich gehöriger massen in die Ahren kaum angesetzet, dahero Theurung entstanden [...]* | The great long-lasting drought arrived on the St. John's Day [24.06.]; the ground became dry, the crops became dry; flax and barley grew very poorly before the proper ear of grain had come out, which caused very high prices […] | Gomolcke, p. 54 |

Severe (strong) droughts that lasted almost the whole season but no longer (up to about 2–3 months) were marked with the -2 index. When they fell in the spring period of plant growth, they influenced the quality of the harvest. It was frequently reported that crops had dried up in fields on hillslopes especially exposed to the sun and with less humid soils than in the valleys. Those droughts made it difficult for people and animals to obtain water; sometimes they prevented the work of some mills on the rivers, but they did not paralyse grain milling in the entire province. Droughts were incidentally related to forests and meadows. Efforts were made to focus on those descriptions in which at least two of the phenomena described above appeared. There was no requirement to describe such droughts in more sources. Examples of descriptions of severe droughts in different historical sources are presented in Table 4.

Table 4. Examples of descriptions of severe droughts in 15th–17th-century sources

| Year | Description | Translation | Source |
|---|---|---|---|
| 1456 | *Fuitque anno eodem precipue circa partes nostras, ubi plures sunt agri sabulosi et argillosi, post festa paschalia siccitas magna et usque ad messem continuata. Messis autem tante humiditatis et instabilitatis,* | And that year there was an exceptionally great drought in our area, where there are numerous sandy and loamy soils; it occurred after the Easter holidays and lasted until the harvest. In the harvest | Catalogus abbatum Saganensium, in: Scriptores rerum Silesiacarum, vol. I, p. 340 |



| | | | |
|---|---|---|---|
| | | period it [the weather] was so wet and unstable [...] | |
| 1472 | *Dieser Sommer, von Pfingsten bis auf aller Heiligen, war ganz trocken und warm* [...] | That summer from Whitsunday to the All Saints Day it was quite dry and warm [...] | Pol, vol. 2, p. 89 |
| 1532 | *Ein dürrer Sommer. Es regnete in sieben Wochen nicht. Das Getreide und die Weide verdorrete auf den Hügeln ganz aus. In etlichen Dörfern war kein gar Wasser. Auf dem Lande konnte man nicht mahlen. Zu 10. 12. 18. Meilen musste man zur Mühle führen. Die Olau trocknete und dorrete auch aus, und hatte kein Wasser bis auf Bartholomei [24.08].* | Dry summer. It did not rain for seven weeks. The grain and grass on the hillsides dried up. In some places there was almost no water. In the countryside, it was impossible to grind grain. One needed to go 10, 12, 18 miles to reach mills. The Oława River dried up [Silesia, auth. suppl.] and there was no water in it until the Saint Bartholomew's Day [August 24]. | Pol, vol. 3, p. 72 |
| 1585 | *Mensis hic [March] fuit serenissimus usque ad miraculum et siccus* | That month [March] the weather was fine and it was dry | Reszka, p. 91 |
| 1637 | *Przy przeważającej w tym miesiącu suszy ogień zniszczył liczne miasta i wsie, widać słabnące plony* [...] | With the drought that prevailed that month, fire destroyed many cities and villages, we could see the yields failing [...]. | Radziwiłł Albrycht Stanisław, Pamiętnik o dziejach w Polsce, vol. 2 1637–1646, A. Przyboś, R. Żelewski (eds), Warszawa, 1980 |
| 1665 | *Der Sommer des Jahres 1665 wird als ungemein heiss angegehen, und soll es die ganzen Hundstage (10.07.–* | The summer of 1665 was incredibly hot; not even once did it rain – so called "Dog Days". | Wernicke, Gesch. Thorns., vol. 2, p. 321 |



| | | | |
|---|---|---|---|
| *20.08) hindurch auch nicht einmal geregnet haben.* | | | |

Moderate droughts – marked with the -1 index – were ones whose appearance was noticeable by people; however, they lasted for a relatively short period of time and affected crops to a limited extent. This group also includes records that seem incidental, are not confirmed in other sources, or may indicate a small range of drought, yet they were significant enough to be recorded in the sources (Table 5). There is no record of consequences (including economic ones). In the description of the drought, a superlative adjective is not used. There appear such expressions as "*dürrer Sommer*" [dry summer]. In other sources, in reference to the same period of time, there may be records that indicate, for example, rain instead of drought.

Table 5. Descriptions of moderate droughts in 15[th]–17[th]-century sources

| Year | Description | Translation | Source |
|---|---|---|---|
| 1461 | *Eodem anno fuit estas calidissima et fluvius Odere valde modicus, similiter et alii fluvii.* | That year the summer was the hottest and the water level of the Odra River fell, as did other rivers'. | Sigismundi Rosiczii chronica, p. 78. |
| 1531 | *Nazajutrz po bitwie pod Obertynem kometa nie dała się iuż tak świetnie widzieć iako przesłey nocy: która ieśli nie porażkę Wołoską, tedy suszą podobno znamieniowała; iakoż tego czasu była susza wielka.* | The following day, after the battle of Obertyn, the comet did not let itself be seen so well as it had the previous night, which augured the defeat of the Vlachs, or drought; And then the drought was really great. | Bielski, p. 311 |
| 1552 | *Den 5 Junii [...] nach der Vesper und grosser Dürre kam ein gewünschter Regen, aber mit grossem Wetter* | On June 5 [...] after the evening and after a great drought, came the desired rain with a great storm. | Pol, vol. 3, p. 158. |
| 1661 | *Es folgte aber ein dürrer Sommer.* | However, a dry summer came. | Happelius, p. 148. |

Therefore, relatively long periods of fifty years were adopted to assess long-term (secular) frequencies. It should also be added that most probably in the oldest sources from the 15[th]–17[th] centuries, primarily droughts of considerable intensity were recorded (i.e. droughts referred to by us as severe and extreme), while those of a smaller scale (moderate) were omitted. This is due to



the nature of the sources at the time and the relatively modest number of such records. Therefore,
it can be assumed that droughts of -1, and probably in some part also droughts of -2 may be
underestimated from the perspective of historical sources. The sources for the 18th century are
definitely more precise. In the 18th century, the duration of the drought as well as its territorial
range can often be very precisely determined, though not always.
**3.2 Dendrochronological data**
We hypothesised that narrow tree rings are linked with drought. The limited access to water during
the vegetation season leads to a water deficit in trees and as a consequence the cambium activity
decreases and produces fewer cells, which is  positively correlated with tree-ring widths (Liang et
al., 2013). Analyses by means of specific packages mentioned below means that we used packages
in the R program (R Development Core Team, 2007). At first the relationships between tree growth
and precipitation was checked. We analysed the effect of climate monthly precipitation and
temperature on tree-ring widths using the treeclim package (Zang and Biondi, 2015). Analysis of
climate growth relationships for monthly data for Toruń revealed that precipitation during the
vegetation season plays a significant role for both pine and oak. A significant positive correlation
was observed for June and July for pine, while for oak a positive correlation was observed for the
previous August and current June and a negative correlation for August (Fig. 2).

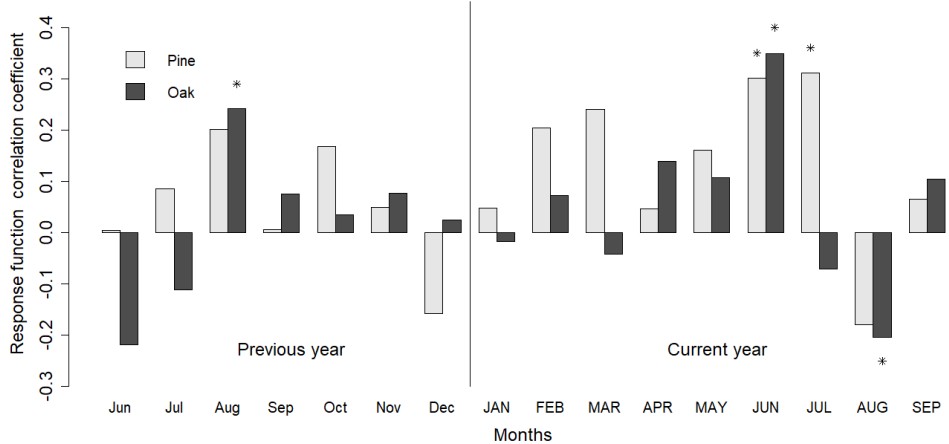

Fig. 2. Climate growth relationships between tree rings in pine (grey bars) and oak (black bars)
and monthly totals of precipitation. Key: Asterisks indicate statistically significant correlation
coefficients at the level of 0.05.





Next we used daily data for Toruń and tree-ring chronologies of pine and oak representing Region
III. According to previous findings, the climate growth relationships are comparable at different
sites in Poland (Zielski et al., 2010), so we used the relationships between daily data and Site 11
and 13 as a model for the rest of our study sites. The reason for this generalisation was also the
limited access to daily data. A period of 90 days for the years 1947–2015 was used to find the
significant relationships between the daily precipitation data and indexes of tree rings. For this
purpose we used the dendroTools package (Jevšenak and Levanič, 2018). The optimal window of
days was revealed to be from May 6 to August 3 for pine, and from April 21 to July 19 for oak.
Study of climate-growth relationships with daily precipitation data from 1947 to 2015 for a 90-
day optimal window width revealed optimal selection from May 6 to August 3 for pine, and from
April 21 to July 19 for oak. The sums of daily precipitation for these periods were summed and
correlated with indexed growth in years of growth reduction (narrow rings) and growth recovery
(wide rings). The correlation coefficient is 0.79 (p<0.05) for pine, and 0.65 (p<0.05) for oak. Next,
the same summed daily precipitations for the selected periods were correlated with the remaining
tree ring indexes (after exclusion of wide and narrow ring indexes). The correlation coefficient is
0.40 for pine and 0.16 for oak.

17        To determine the pointer years we used the dplR package (Bunn, 2008). The minimum

absolute relative radial growth variation, above which the growth change from year $t$-1 to $t$ is
considered significant, was 10. Any year in which more than 95% of trees per site displayed
significant relative radial growth variations above 10 was qualified as "extreme reduction"; "great
reduction" was determined as between 85–95% of trees; and "moderate reduction" was between
75% and 85%.

24        **2.3 Instrumental data**

As results from Table 2, for the analysis of droughts in the instrumental period, eight long-term
series of monthly totals of precipitation have been used. All these precipitation series were checked
for completeness. The few data gaps in the analysed series were completed using homogenised
precipitation series from the nearest stations. For this purpose, a simple method of constant
quotients was utilised (Pruchnicki, 1987). However, due to the lack of available reference series,
such a procedure was not used to fill data for the period 1880–1884 for Orzysz. Homogenisation
of all the used precipitation series was checked using the AnClim software (Štěpánek et al., 2009).

32        On the basis of the completed series of atmospheric precipitation, the possibility of

obtaining a synthetic precipitation index for the whole country was tested. A similar method was
adopted in Brázdil et al. (2007) to determine drought indices in the Czech Republic for the period
1881–2006. In Poland, Kożuchowski (1985) presented a 100-year series of average areal annual
atmospheric precipitation for 1881–1980 (his Table 3) calculated from data from 12





meteorological stations using precipitation regression equations relative to altitude above sea level.
Miętus (1996), in turn, presented mean areal precipitation for the Coast area. For the analysis, we
took 30-year moving correlation coefficients ($r$) for monthly totals of precipitation counted for the
period 1901–2000. All correlation coefficients were statistically significant ($p<0.05$) with values
varying from 0.46 to 0.71 (see Table 6, upper part). Only the Kraków series had a significantly
lower value of $r$ (the highest value of 0.33 described the relationship between Kraków and
Wrocław). For annual precipitation totals in the period 1951–2000, Kożuchowski and Żmudzka
(2003) obtained only slightly higher values of correlation coefficients, varying from 0.6 to 0.8.
Unsatisfactory results of $r$, particularly related to the series for Kraków, suggested that we should
not construct monthly precipitation series for Poland. It seems that the number of long-term
precipitation series is probably relatively too small for a country of such area (312,679 km$^2$).
Further analysis was thus carried out on regions delimited by a landscape criterion, though this
excludes mountains, whose atmospheric precipitation is spatially and temporally far more variable
(Kożuchowski, 1985).
Table 6. Correlation coefficients between monthly totals of atmospheric precipitation
(upper part of table) and SPI1 (lower part of table) in area of Poland calculated based on data from
the period 1901–2000

| Station | Toruń | Koszalin | Gdańsk | Orzysz-Mikołajki | Poznań | Warszawa | Żagań-Wrocław | Kraków |
|---|---|---|---|---|---|---|---|---|
| Toruń | | *0.56* | *0.67* | *0.62* | *0.69* | *0.62* | *0.61* | *0.29* |
| Koszalin | *0.56* | | *0.71* | *0.55* | *0.55* | *0.52* | *0.46* | *0.20* |
| Gdańsk | *0.62* | *0.69* | | *0.66* | *0.58* | *0.61* | *0.55* | *0.26* |
| Orzysz-Mikołajki | *0.55* | *0.53* | *0.60* | | *0.55* | *0.71* | *0.54* | *0.31* |
| Poznań | *0.66* | *0.57* | *0.55* | *0.49* | | *0.58* | *0.68* | *0.25* |
| Warszawa | *0.58* | *0.48* | *0.52* | *0.63* | *0.53* | | *0.61* | *0.28* |
| Żagań-Wrocław | *0.56* | *0.44* | *0.47* | *0.45* | *0.64* | *0.53* | | *0.33* |
| Kraków | 0.00 | -0.03 | -0.03 | -0.03 | -0.03 | -0.02 | 0.00 | |

*values statistically significant at the level of p<0.05 are shown in italic*

The aim of analysis of instrumental series was to calculate the number, length and category
of droughts in the area of Poland since 1722, i.e. for almost 300 years. The Standardised
Precipitation Index (SPI, McKee et al., 1993) was calculated from monthly precipitation totals to
explore the occurrence of droughts in the analysed locations (Table 2). This index is one of the
simplest methods used to identify meteorological droughts, since it uses only monthly totals of
precipitation and is therefore widely used in the literature. Osuch et al. (2015) state that the SPI is
used for both research and operational purposes in over 60 countries. The SPI index is also most
popularly used in Poland (e.g. Łabędzki, 2007; Kalbarczyk, 2010; Bąk et al., 2012; Bartczak et al.,
2014; Osuch et al., 2015, 2016; Bąk and Kubiak-Wójcicka, 2017). What is more, the SPI is used
also by two institutes mentioned in Section 1 (IMGW-PIB, and the Institute of Technology and





Life Sciences [ITP]) and also by the Institute of Soil Science and Plant Cultivation, which is
responsible for agricultural drought monitoring in Poland (for more details see Łabędzki and Bąk,
2014). Hence our decision to also use this index in our work.

4        The program SPI Generator (National Drought Mitigation Center, University of Nebraska),

was used to perform this analysis. SPI was initially calculated for 1-, 3- 6-, 12- and 24-month time
scales. Further analysis was, however, done using SPI calculated only for 1-, 3- and 24-month time
scales. All of them represent meteorological droughts, from short-term to long-term, respectively.
The last two (SPI3 and SPI24) can also be used as a good proxy for agricultural and hydrological
droughts, respectively. For climate conditions in Poland it was shown that there exists a strong
spatial relationship of SPI values (Table 6, lower part). Significant empirical relations were also
found between SPI and pure agricultural and hydrological indices. Łabędzki et al. (2008) found
high correlation coefficients ($|r|>0.7$) between SPI and some agricultural indices such as: crop
drought index (CDI), water deficit (N) and relative duration of soil moisture deficit ($t_{def}$.). On the
other hand, a much weaker relation ($r< 0.5$) was found between SPI24 and hydrological droughts
estimated based on SWI-24 (24-month standardised water level index) for the Vistula river in
Toruń by Bąk and Kubiak-Wójcicka (2017). According to them, this relation was reduced by the
influence of external factors (the hydropower plant in Włocławek, major groundwater basin), and
climate factors appearing in the upper and middle part of the river basin.

19       To identify droughts (dry months), the criterion proposed by McKee (1993) and modified

for Polish climate conditions by Łabędzki (2007) was used. Droughts were divided into three
categories based on SPI values: moderate droughts (-0.50 to -1.49), severe (-1.50 to -1.99), and
extreme (≤-2.00). Methods that identify multi-month droughts using the SPI calculated for
different, rigidly defined numbers of consecutive months (3, 6, 12 or 24) simplify analysis,
especially in terms of drought duration and calculating the cumulative intensity of the whole
phenomenon. Therefore, in this work, we have adopted the following criteria to identify droughts
and determine their duration. Firstly, instances of an SPI1 value within any of the above ranges
for only a single month were considered irrelevant. Secondly, a drought was considered to be at
least two consecutive months during which the SPI1 value was ≤-0.50. Thus identified, a drought
was determined both in terms of duration and by category. Thirdly, drought category was
determined by the dry month of lowest SPI1 value. A drought was thus considered extreme if the
SPI1 value for at least one of the drought months was ≤-2.00. If the SPI1 of the driest month within
a particular instance of drought was between -1.50 and -1.99, the drought was determined to be
severe. The remaining droughts were qualified as moderate. Number of droughts was determined
for years and for climatological seasons. A drought's final month determined its season.

35       Drought is a widely occurring phenomenon, but its frequency is extremely limited within

particular long-term periods. For this reason, it was decided to group numbers of droughts into





longer periods. For a fuller comparison of drought occurrence identified on the basis of
dendrochronological data (narrow rings), we used instrumental data to calculate the number and
duration of droughts within ten-year periods, starting from the slightly shorter period 1722–1730,
through full decades, to the five-year period 2011–2015. Next, we also summed the number of
droughts by 50-year period, also determining seasons in this case, just as we did when analysing
the documentary data.
For the purpose of comparison of SPI1 values (meteorological droughts) against historical
indices (-1, -2 and -3) the following assumptions were established: the -1 index was attributed to
SPI1 values ranging from -0.50 to -1.49; -2 for the range -1.50 to -1.99; and -3 for SPI1 ≤-2.00.
Frequency of occurrence of meteorological droughts for the instrumental period was calculated
for standard meteorological seasons (Dec–Feb, Mar–May, etc.) as well as for May–July. This
allowed for comparison of the occurrence of droughts against their statistics available in
documentary evidence (seasons) and dendrochronological data (May–July). The last period was
added because for this time a significant influence of precipitation on tree-ring widths in Poland
was found (see Sect. *Methods*). It was revealed that most of the growth reduction (negative pointer
years) was related to the occurrence of drought. Thus, years with extreme, great and moderate tree
growth reductions can roughly, and with a large probability, indicate the occurrence of extreme,
severe and moderate droughts, respectively. In the case of documentary data such droughts were
described using indices -3, -2 and -1.
As mentioned in Section 3.1, information about droughts in historical times is rather
heavily underestimated, in particular in the case of moderate droughts, and therefore documentary
identified droughts of categories -2 and -3 have frequently been used for the purpose of comparison
against other sources. Such an approach also increases the probability that identified droughts
occurred in large part of Poland. In addition, to be sure that they were caused only by climate, the
assumption of their occurrence in minimum two geographical regions was usually also utilised.
On the other hand, for comparison of droughts delimited using dendrochronological and
instrumental data, all categories of them were used.
The number of months $N_i$ in each class of drought intensity (moderate, severe and extreme)
was computed for the 1- 3-, and 24-month timescales. Then the number of droughts per 100 years
was calculated according to the following formula proposed by Łabędzki (2007):
$$N_{i,100} = \frac{N_i}{i \cdot n} \cdot 100$$
where:
$N_{i,100}$ – the number of droughts for a timescale $i$ in 100 years



$N_i$ – the number of months with droughts for a timescale $i$ in the $n$-year set
$i$ – timescale (1, 3, 24, months)
$n$ – the number of years in the particular study data set
**4.  Results**
**4.1 Droughts in Poland based on documentary data**
It seems that droughts were not very frequent in Poland. In particular regions (including droughts
presented in sources as nationwide, and therefore also noticeable in individual regions) in total
from 33 to 71 droughts were recorded between 1451 and the end of the 18[th] century (Fig. 3). Most
of those were recorded in Pomerania and Silesia, and the least in Greater Poland, Masuria and
Mazovia (Figs 3 and 4). This is undoubtedly not a reflection of the frequency of droughts in
individual regions, but a consequence of the sources preserved for each region. Without a shadow
of doubt, the richest and most accurate sources come from two regions: Pomerania ( especially
from big cities like Gdańsk, Toruń and Elbląg) and Silesia. It very often happens that one drought
is described in many sources from the region; moreover, it is confirmed by records referring to the
entire territory of Poland. A drought described in this way can be analysed more accurately. The
sources from Greater Poland, Mazovia and Masuria are definitely poorer. Consequently, it is
probable that the number of droughts in these regions was actually higher, and close to the number
of droughts in Silesia or Pomerania.
Information that refers to the same year and comes from different regions confirms a larger
territorial range of drought. This does not mean, however, that in cases where such information
was preserved only for one of the regions, other areas were not affected by drought. This lack of
reports may have resulted from the lack of appropriate sources, and not from the fact that there
was no drought in a given region. These numbers undoubtedly depend on the surviving sources
and reflect part of the actual state of affairs. In order to partially compensate for these source
deficiencies, it was assumed that the records referring to drought in the whole country refer
simultaneously to each of the six identified regions.





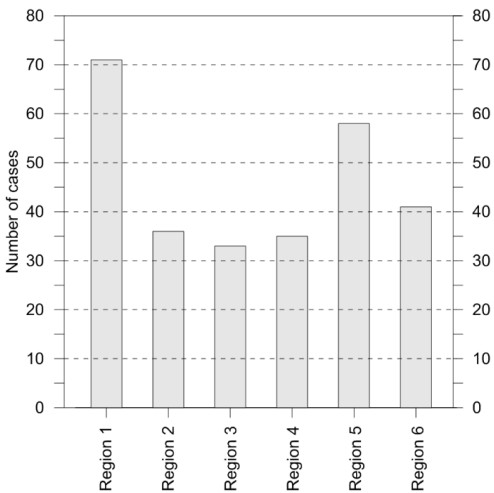

Fig. 3. Number of years with droughts in six geographical regions of Poland (including information
related to the whole country) 1451–1800. See Table 2 or Fig. 4 for names of regions.

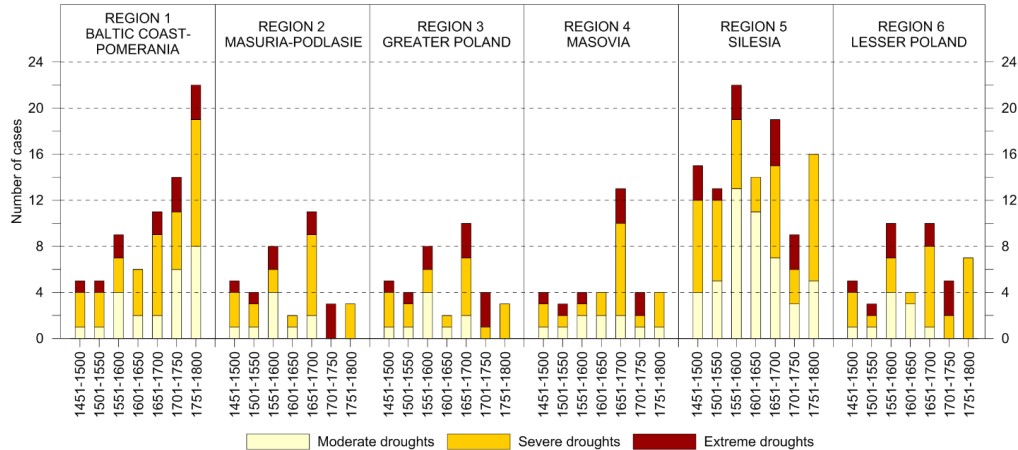

6        Fig. 4. Frequency of occurrence of three categories of droughts in six distinguished

geographical regions in Poland in 50-year periods, 1451–1800

9        We also calculated the frequency of droughts covering a large part of Poland, i.e. more than

one region (Fig. 5). In the chronological order in the periods of 50 years, the number of extreme
droughts (-3) never exceeded five; in the first half of the 16th century only the drought of 1549 was
recognised as such, while in the first half of the 17th century, extreme droughts were completely
absent (Fig. 5). It seems that extreme droughts, whose total number in the period 1451–1800 was
17, were regularly recorded in sources, and this information is quite reliable.





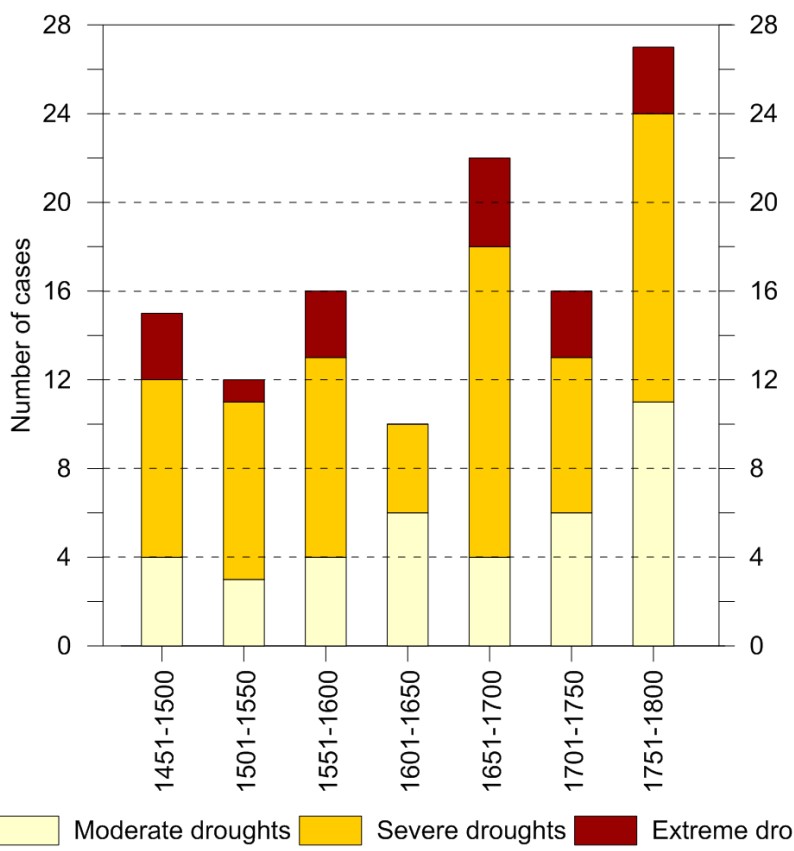

Fig. 5. Frequency of occurrence of three categories of droughts in large part of Poland in 50-year periods, 1451–1800

The number of severe droughts (-2) was usually between four and nine in particular periods of fifty years. Many more droughts belonging to this category were recorded in the second half of the 17th century and in the second half of the 18th century; their numbers were respectively 14 and 13 (Fig. 5).

However, the total frequency of extreme (-3) and severe (-2) droughts amounted to 80 and ranged from 4 to 12 in particular fifty-year periods, except for the second half of the 17th century and the second half of the 18th century, when there occurred as many as 18 and 16 droughts, respectively (Fig. 5). The increase in the number of identified droughts in the second half of the 17th century was certainly due to the availability of detailed weather records from the period 1656–1685 taken from the memoirs of Jan Antoni Chrapowicki (Nowosad et al., 2007). However, the minimum number of droughts (only 4) took place in the first half of the 17th century (Fig. 5), for which, in turn, we recorded significant losses in the sources.





The number of moderate droughts (-1) varied in all 50-year periods from 3 to 6, except for
the second half of the 18[th] century, when there were recorded as many as 11 droughts belonging
to this category (Fig. 5). A larger number of such droughts starting from the beginning of the 18[th]
century undoubtedly results from regional sources being more accurate. In this century, many
historical sources were created; they now allow for a fairly accurate reconstruction of the weather
condition, including the appearance of smaller droughts and prolonged shortages of rainfall.
Spring (31) and summer (37) droughts prevailed among the recorded droughts. Also,
droughts in spring–summer were often mentioned (22), but much less frequently in summer and
autumn (4). Rare were droughts that occurred only in autumn (4). Winter droughts were reported
only in three years. In the case of many reports mentioning "a drought occurring this year" it is
difficult to decide what the time of its occurrence was.
Nevertheless, the findings should be treated with some caution. The specificity of the
chronicle's narrative was that weather phenomena were recorded in the case of their extreme rare
character, or because of their consequences for human existence. Droughts undoubtedly posed a
serious threat to crops during periods of plant growth – above all in spring and summer. In the case
of winters, the lack of snowfall could hardly be perceived as a manifestation of drought.
**4.2 Droughts in Poland based on dendrochronological data**
Twenty-two local chronologies of trees (pine, oak, and fir) from Poland were taken into account
for detecting negative pointer years, showing narrow rings. In a year in which we have narrow
rings at more than 1 site, we count this pointer year as a "multiple observation" year, whereas, in
a year with only one observation, at one site, we call it a year "without multiple observation". In
total, 758 pointer years with multiple observations were detected and 432 years without multiple
observations. There are 237 multiple observation years of extreme reduction, 122 of great
reduction, 252 of moderate reduction and 147 negative pointer years from the literature (Opała and
Mendecki, 2014; Opała, 2015; Szychowska-Krąpiec, 2010) (Fig. 6). The number of pointer years
in selected 50-year periods varies (Fig. 7) and is at least 30 within the years 1401–1450 and within
each of the 50-year intervals from 1701 to 1950. The evidently smallest number of negative pointer
years occurred in the first 150 years (Fig. 7). In the years 996–1000, drought did not occur, and
therefore this period was omitted in Figures 6 and 7. The number of chronologies varies and
depends on region. More chronologies in the last 300 years result from existing old trees. It also
led to the detection of more pointer years. According to Neuwirth et al. (2007) during extreme
climatic conditions trees react in the same way, but during years of less pronounced weather conditions
regional differences in growth reactions increase. Narrow rings observed in the same year in trees from
different regions suggests extreme climatic conditions.







**1001-1340**

**1341-1680**

**1681-2015**

Percent of samples with growth reduction
or pointer years according to the literature

| more than 95% | less than 85% |
| from 85% to 95% | droughts according to various authors |





Fig. 6. Pointer years in Poland, 1001–2015

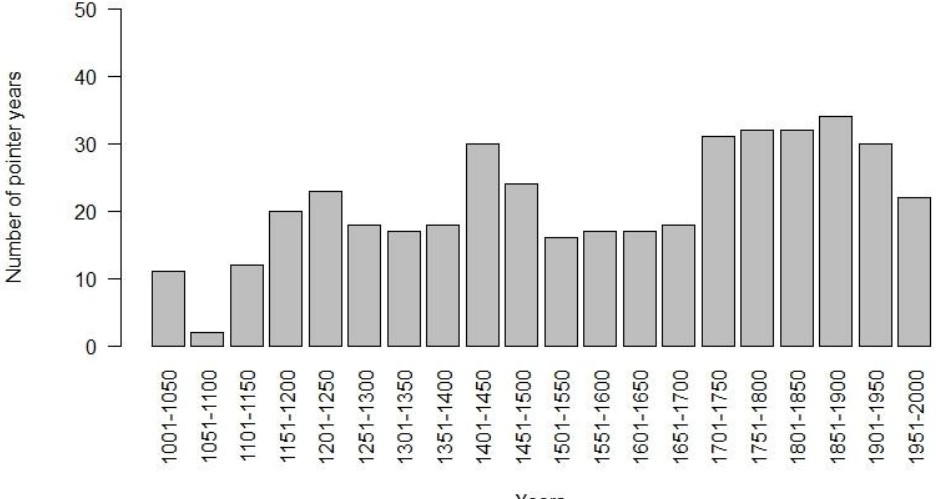

Fig. 7. Number of negative pointer years (without multiply observation – i.e. narrow
rings in 1976 were observed on six samples but are treated as a one-pointer year) in
Poland in 50-year periods, 1001–2000
**4.3 Droughts in Poland based on instrumental data**
Instrumental observations of precipitation in Poland are among the longest-standing in the world
(Filipiak 2007). As results from Table 2, they are available since 1722. In Figure 8 we present the
SPI calculated for eight sites in Poland for 1-, 3-, and 24-month time scales. The values of SPI3
and SPI24 were filtered by 10-element and 30-element low-pass Gauss filters, respectively, in
order to more clearly distinguish long-term dry periods. The analysis of Figure 8 reveals that the
occurrence of droughts in different areas of Poland shows both similarities and discrepancies. It is
very clear that in northern and central Poland, a long-term (24 months' duration, red line) and
extreme drought occurred at the threshold of the 1850s/60s. Almost one hundred later (at the
threshold of the 1940s/50s) such a strong drought was present across the entire area of Poland (Fig.
8). Except for Kraków, and also Gdańsk in the last few years, severe droughts have not been
observable at the turn of the 21st century. In Silesia, a very dry period occurred for almost the entire
first half of the 19th century, and then significantly less severe droughts occurred here only in the
1950s and 1990s. For the 18th century we have mainly information for Gdańsk. Figure 8 shows





that dry periods (moderate droughts) occurred here only at the threshold of the 1750s/60s and in
the mid-1770s. The most extreme droughts in different parts of Poland occurred in different times.
For example, in Gdańsk at the threshold of the 1910s/20s, in Koszalin and Orzysz-Mikołajki in
the 1850s, in Toruń in the 1910s, in Poznań in the 1980s, and in Kraków in the 1980s and 1990s
(Fig. 8).







Fig. 8. Variability in SPI: 1-month (grey curve), 3-month (black curve) and 24-month (red curve) calculated from the Polish instrumental series listed in Table 2 (oriented from north to south) in the period 1722–2015. SPI-3 and SPI-24 were filtered by 10-element and 30-element low-pass Gauss filters, respectively.

Trends calculated for three types of SPI (SPI1, SPI3, and SPI24) are very small and not statistically significant in all study regions. This means that long-term frequency of droughts in Poland has been stable for the last two or three centuries.

The number of moderate, severe, extreme and all-category droughts (see Section *Methods* for definitions) in ten-year periods calculated from the Polish instrumental series listed in Table 2 (oriented from north to south) in the period 1722–2015 is presented in Figure 9. In the period 1876–2015, for which complete series of SPI are available for all study sites, the number of all-category droughts (Fig. 9D) varies mainly in the ranges 3–4 and 8–12 per decade. Below the lower threshold of this range we must mention the occurrence of only two droughts in the decade 2001–2010 in Warszawa. On the other hand, this range of frequency was exceeded in only three decades. The greatest 10-year number of all-category droughts (14) in the study period was noted in Gdańsk in the decade 1881–1890. In another two decades (1951–1960 and 1991–2000) 13 droughts occurred in Toruń and Kraków, respectively (Fig. 9D). Two decades 1851–1860 and 1861–1870 were very dry in Poland, in particular in its northern and western parts, and the number of droughts varied between 6 and 10 per decade. For pre-1850, the information about drought occurrence is significantly sparser, but it can be stated that in both areas for which data exist (Silesia and Masovia) the number of droughts in the first half of the 19[th] century (8–14 per decade) was higher than in the rest of the study period. The contrast is particularly great for Silesia (see also Fig. 8). For the 18[th] century we only have information for Gdańsk. Figure 9 shows that their number in this time (from 4 to 8–9 per decade) was typical of the rest of the study period.







Fig. 9. Decadal frequency of droughts in Poland in 1722–2015 identified using SPI1

Key: A – moderate droughts, B – severe droughts, C – extreme droughts, D – all-category droughts

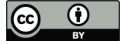



In line with expectations, moderate droughts evidently dominate, usually with a frequency
of 2–8 per decade (Fig. 9A), then severe (Fig. 9b), and extreme (Fig. 9c) with typical frequencies
not being much different, at 1–4 per decade and 1–3 per decade, respectively. In terms of these
drought characteristics, as with the characteristics described by SPI1, SPI3 and SPI24, no long-
term trends are observable in Poland for the last two or three centuries (Fig. 9).
For comparison against the number of droughts delimited using documentary evidence, 50-
year frequencies of the three categories of droughts were calculated for climatological seasons
(Fig. 10). It comes as little surprise that the frequency of all-category droughts was greatest in
winter. Other seasons show more-or-less similar frequencies. In winter, droughts evidently
dominated in the study period in the second half of the 19$^{th}$ century, this is particularly well seen
in the case of severe droughts, and slightly less so for moderate droughts, which were also quite
frequent in the first half of the 20$^{th}$ century. Extreme droughts in winter do not show any significant
changes over time, but it should be emphasised here that they were slightly more frequent in 1951–
2000 than in 1851–1900. In spring, moderate droughts prevailed still in the period 1851–1950
(usually 4–6 cases), with a greater frequency in the first half of the period. Both severe and extreme
droughts were most frequent (usually 1–3 cases) in 1851–1900, and in particular in 1951–2000
(Fig. 10). In summer there is a clear change in the time pattern of drought occurrence: drought
frequency rises in the 20$^{th}$ century, and in the case of moderate droughts particularly in its second
half. The contrast in drought frequency between the 20$^{th}$ century compared to pre-1900 is very
clear, primarily in the case of extreme droughts. In autumn, moderate droughts do not show
changes in the last two centuries, while severe and extreme droughts were most frequent in the
first and second halves of the 20$^{th}$ century, respectively (Fig. 10).

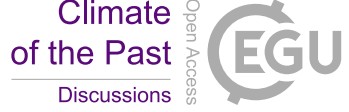



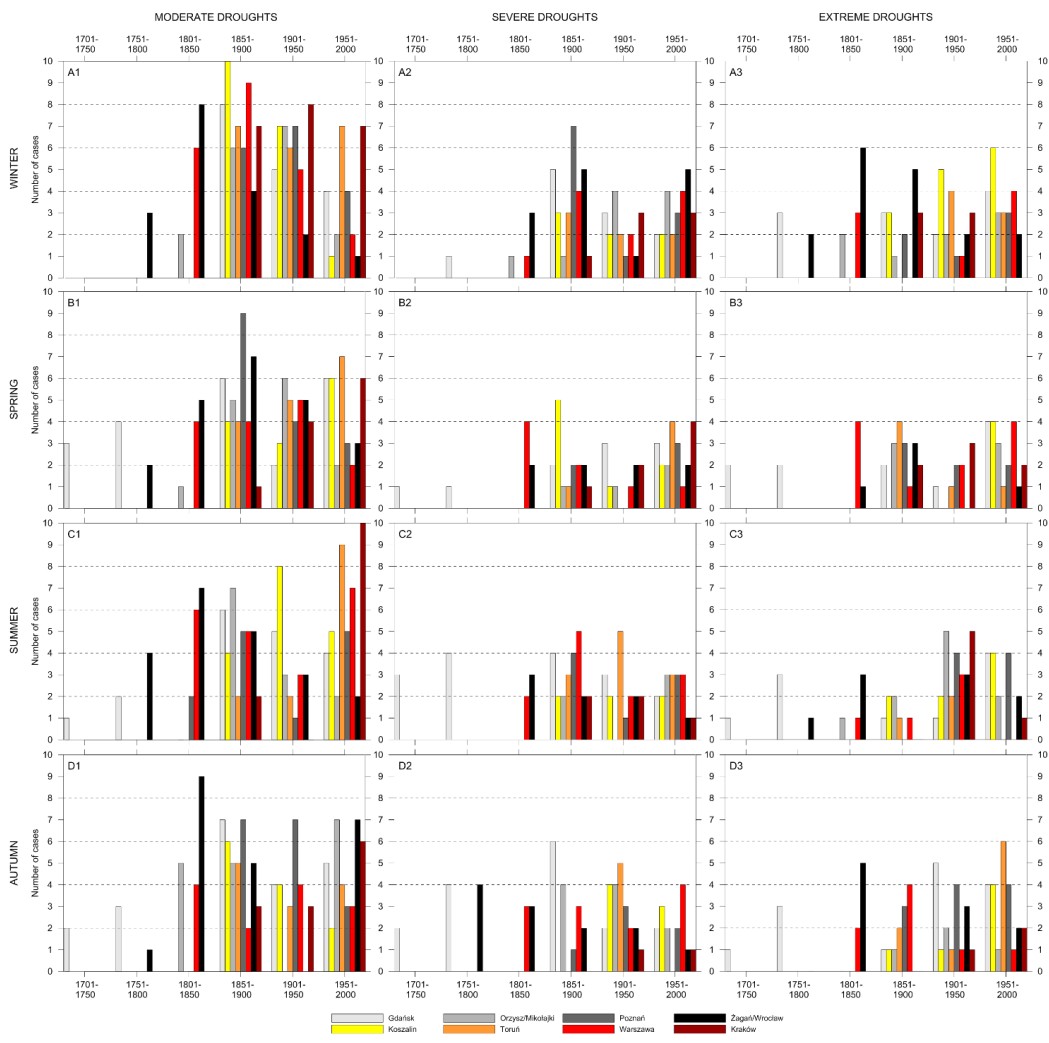

Fig. 10. Seasonal 50-year frequency of droughts in Poland in 1722–2015 identified using SPI1

The frequency of droughts per 100 years and their duration is shown in Figure 11. The greatest number of all-category droughts occurred in Gdańsk (165) and in Żagań/Wrocław (155), while the smallest was in Kraków (104). In line with expectations, moderate droughts clearly dominate (55–75). The number of severe and extreme droughts is more-or-less comparable, most often ranging between 25 and 40. Both these two categories of droughts were most frequent in the coastal part of Poland, and least frequent in Lesser Poland (Fig. 11). Most droughts lasted two months (about 60–70%), and then 3–4 months (10–20%). The frequency of droughts of 5-or-more months was less than 10%. The longest droughts had durations of 7–8 months and occurred in Gdańsk from January to July of 1771, in Wrocław from March to September of 1805, in Poznań





1   from May to November of 1874, in Toruń from March to September of 1900, and in Wrocław

2   (again) from August 1953 to March of 1954 (8 months).

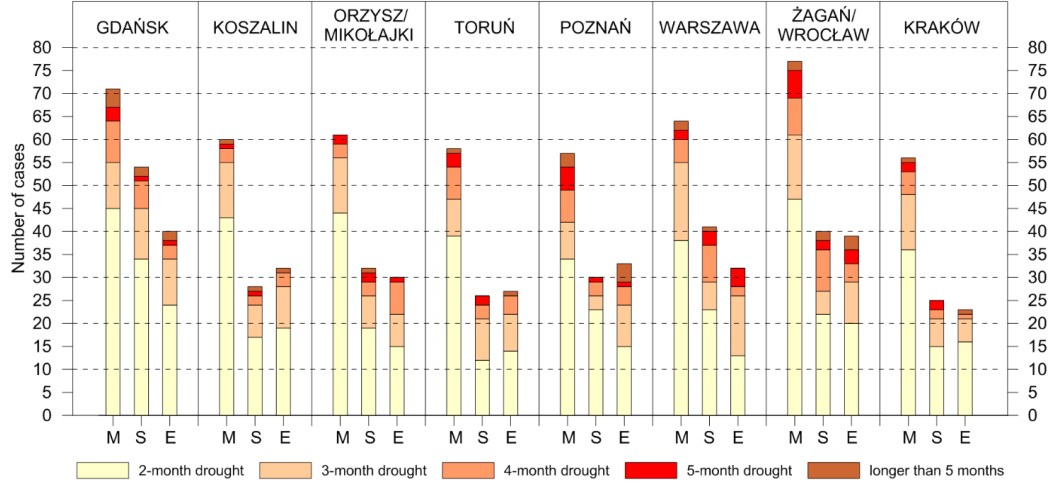

6       Fig. 11. Average frequency of three categories of droughts (M – moderate, S – severe, E –

extreme) in Poland per 100 years stratified by duration, 1722–2015

9       Łabędzki (2007) proposed a simple formula to calculate the frequency of occurrence of dry

months and droughts per 100 years based on SPI values (see methods). Using his formula we
calculated frequencies of dry months using SPI1, short-term droughts (SPI3) and long-term
droughts (SPI24), including three categories of them (see Fig. 12). Analysis of this figure shows
that the number of dry months in Poland usually ranges around 350 per 100 years (from 342 in
Orzysz/Mikołajki to 366 in Poznań). The number of short-term droughts (SPI3) for Poland as a
whole is comparable and usually ranges around 120 per 100 years (from 119 in Koszalin to 127 in
Wrocław and Kraków), while the frequency of long-term droughts (SPI24) is 15–16 per 100 years.
The short-term droughts distinguished here using SPI3 are most comparable to droughts delimited
using the method proposed in the paper. Ratios of frequencies between moderate, severe and
extreme droughts are generally similar in both methods (Figs 11 and 12), although in the Łabędzki
method there is a greater domination of moderate droughts over the other two categories. Severe
droughts are also clearly more numerous than extreme droughts (Fig. 12), which is not so clearly
visible in drought frequencies calculated using our method (Fig. 11).



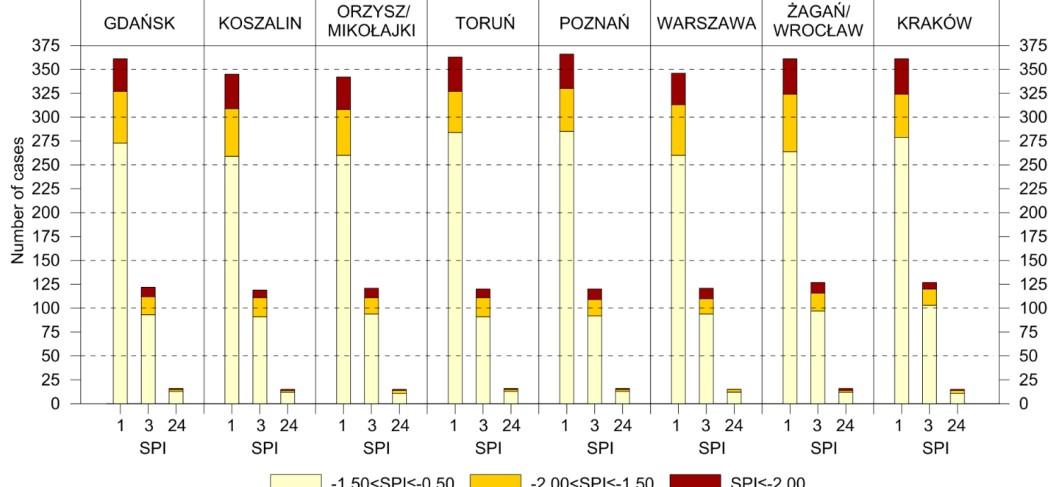

Fig. 12. Frequencies of dry months (SPI1), short-term droughts (SPI3) and long-term droughts
(SPI24) in Poland, including three intensity categories calculated using Łabędzki's formula

### 4.4 Selected megadroughts in Poland from historical times

Based on detailed analysis of all documentary evidence gathered for the period 1451–1800 we
distinguished 17 megadroughts (also referred to in the paper as "extreme droughts", index -3) in
Poland (see Fig. 5). Six of them – the most severe (Fig. 13) – have been chosen for more detailed
presentation here. The main features of each megadrought are described (e.g. time of occurrence,
duration, geographical area, consequences for nature, socio-economic impact).

#### 4.4.1 The year 1473

This drought affected the whole of Europe. In the case of Poland, it was quite well described by
Jan Długosz in "Annales", as Długosz himself observed its course. He wrote about extraordinary
heat and a prolonged lack of rain. He emphasised the extremely low level of water in the Vistula
River and many other rivers that could be easily waded across. Water reservoirs were completely
dry. The lack of water was marked throughout the whole country. Fires were another commonplace
phenomenon. There were forest fires. Długosz also mentioned economic consequences. Fires
destroyed wild beehives in the forests. Drought destroyed the spring sowing. Animals got sick.
Fires affected such cities as Kraków, Wieliczka, Konin, Bełz, Chełm, Lubomia, Łęczyca,
Sandomierz and others (Długosz Ks.) (see Fig. 13). According to the Silesian chronicler Peter





Eschenloer, the drought lasted from 23 April to 11 November. This chronicler recorded an
extremely low level of water in the Odra River. Water mills could not operate. There was no water
in wells. Even wild animals were affected by the lack of water. Similar information was provided
by another Silesian chronicler, Nicolaus Pol. Meanwhile, the author of *Roczniki głogowskie*,
Kaspar Borgeni, reported that the drought lasted only 10 weeks. However, he provided many
detailed dates in his narrative about the harvest time and their quality; there was no rain from April
4 to September 22, so it should be considered that the drought lasted almost 6 months.

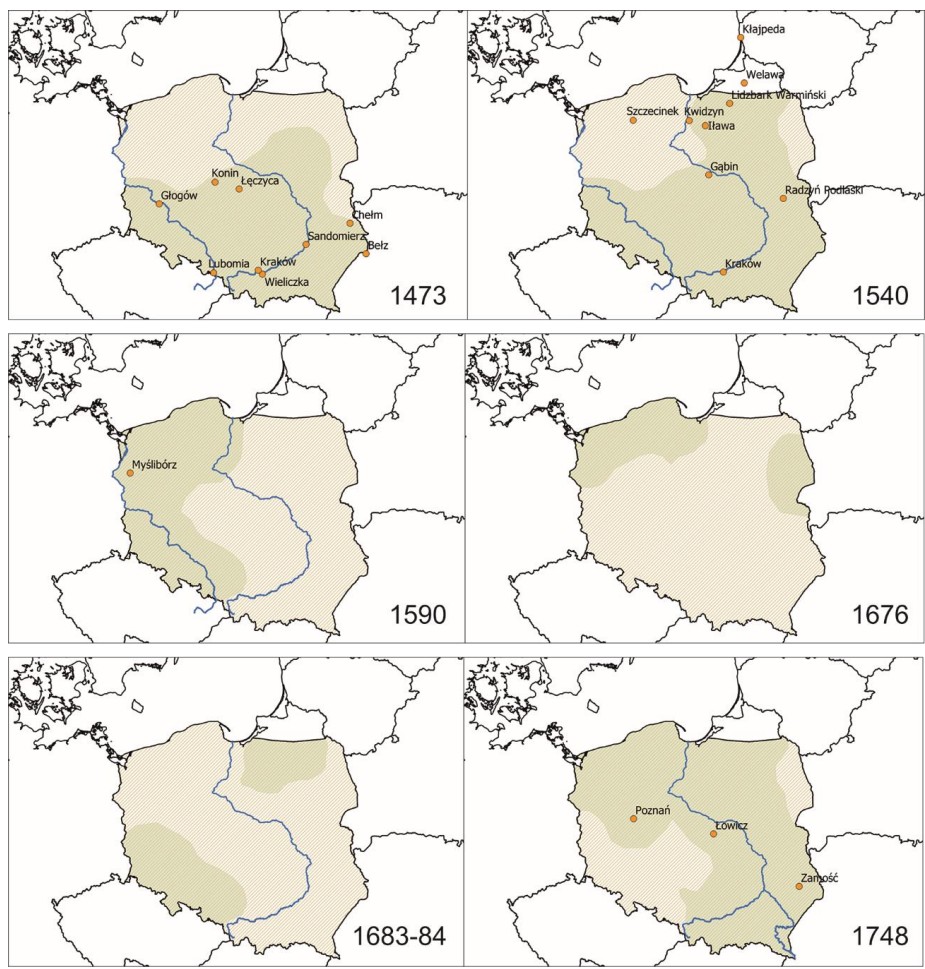

Fig. 13. The most severe megadroughts, with spatial coverage (dark colour). Location of
sites and rivers mentioned in Section 4.4 and Table 3.
4.4.2 The year 1540
This drought belongs to one of the best described droughts in old Europe. In Poland, however, the
year 1540 began with numerous floods in the winter (Poznań) and early spring (Żuławy and
Gdańsk). Heavy rainfalls also caused floods in Świecie. Polish sources are quite laconic, if





unambiguous, about the drought of 1540, considering its scale. A parish priest from Lidzbark
Warmiński wrote about a terrifying drought. The Silesian chronicler Nicolaus Pol wrote about the
drying of many waters and the greening of the Odra River, probably as a result of the development
of algae at high temperatures. It was reported that grass was drying out, cereal harvests were poor,
cattle had to go many miles to watering places. The detailed observations of the Kraków professor
Marcin Biem leave no doubt as to the lack of rainfall and the extreme nature of the drought in the
vicinity of Kraków. The drought lasted until October. There were many fires, including in such
cities as Kwidzyn, Welawa, Klaipeda in the Prussian state, Gąbin in Mazowsze, and Radzyń
Podlaski (see Fig. 13). Fires were also reported in Iława and Szczecinek (Nowosad and Oliński,

10 2019).

12        4.4.3 The year 1590

The winter of 1589/90 was quite harsh, and rivers froze. There must certainly have been spring
thaws. In the literature, mention is made of there having been no rain for 38 weeks. In the vicinity
of Myślibórz (German: Soldin) on 4 May there was a severe frost, followed by a strong heat. There
were also heavy storms. The phenomena resulted in numerous fires. From the end of May there
was an uninterrupted rainless heat that lasted for a very long time. The duration of the heat was
determined to have lasted 38 weeks, which is probably a mistake. Rivers dried up, the river mills
stopped working. Prices rose significantly (Reinhold, 1846, p. 143; Girguś et al., 1965, p. 182).
The dry summer and the drying of many rivers were also mentioned in reference to Silesia and the
Karkonosze Mountanis (Bergemann, 1830a, b). The level of the Vistula River was also extremely
low. The drought therefore affected all Polish areas and lasted continuously from the end of May
to the end of autumn. Many of its manifestations (total lack of rainfall, drying of rivers, high
temperatures, consequences for agriculture and nature) indicate its extreme character.

26        4.4.4 The year 1676

The drought of 1676 was described independently in several sources. Spring is supposed to have
abounded in storms that caused numerous fires. There was drought in the summer. In Pomerania
(see Fig. 13) it rained only twice in the summer. The whole summer was dry and hot. The drought
caused damage to crops in slightly higher areas. The harvest of fruit and vegetables was also poor
due to the drought. In Podlasie, the beginning of January was exceptionally warm, although frosts
arrived later. According to the records from Antoni Chrapowicki's diary, June and July were very
dry months in Podlasie. Chrapowicki wrote that crops "burned out" in the fields. In August and
September, Chrapowicki stayed in eastern Belarus, which is why his records concerning the late



summer and autumn cannot be taken into account (Diaryusz Życia JWJmci Pana Jan Antoni Chrapowicki). The research into the memoirs of Chrapowicki indicates that the precipitation in 1676 was the slightest of all the years covered by his diary (1656–1684) (Przybylak and Marciniak, 2010). In other sources, the high prices that prevailed in the country this year were also underlined (Namaczyńska, 1937).

### 4.4.5 The years 1683–1684

It is known from a rather late record that a great drought was recorded in Masuria in 1683. It caused a lack of crops and high prices. In 1684, after a harsh winter, a hot, dry summer came. The drought resulted in earlier, but thus weaker, harvests of winter grain and the destruction of spring crops. Water reservoirs dried up. There were not enough watering places for animals (Namaczyńska, 1937). According to Silesian sources, the drought came on 24 June; it destroyed grain and flax, and burned grass. Cattle died, for a lack of grass and water. Prices were very high (Gomolcke, 32–33, 54). From various sources it can be established that the drought began at the end of June and continued in July, August and September.

### 4.4.6 The year 1748

The winter was quite long. In Gdańsk, on 7 April, there was ice-floe on the Motława River. In the vicinity of Toruń the ice on the Vistula River did not start to melt until the beginning of April. Near Toruń, the Vistula river flooded adjacent territories. The water level began to fall at the beginning of May. Beautiful, dry weather came, and it started to arouse farmers' anxiety about the growth of plants. On 25 May, it rained in Toruń, but the magnitude of the precipitation was insignificant. The second half of May was considered to be extremely dry. In Gdańsk, heat and drought prevailed from 8 to 23 May. In Toruń, on 7 June an increase was recorded in the water level in the Vistula River, which may indicate more significant rainfalls in the south of Poland. In the vicinity of Toruń, rain fell after a long break, on 11 June, causing people to rejoice, but by 22 June dry weather was again recorded. In Gdańsk, in June, dry days prevailed, but they were interspersed with rainy days.

On 1 July, in Toruń, it was recorded that there had been light rains from time to time, but above all, a great drought had been felt. No fires had broken out in the vicinity yet, but they had in many places in Poland and Lithuania: fires were recorded in Poznań and Zamość (see Fig. 13). In Gdańsk, rainless weather prevailed throughout the first half of July, while in the second half there were only five days with rain. In mid-July, high prices resulting from the prolonged drought were reported. Transport on the Vistula River was extremely difficult due to the low water level.



Information about the drought also came from other European countries. In addition, locusts
appeared in Hungary and Transylvania. In Toruń and Gdańsk, rain fell for a few days after the
solar eclipse of 25 July. Similar rains fell at that time in Warszawa. At the beginning of August,
however, the drought was reported again. In Toruń, rain fell on 5 August, then on 8 August. At
that time, the water level in the Vistula River also increased for a short time, but at the same time,
there were reports of fires having destroyed Łowicz. In Toruń, the drought prevailed until the
end of August and the first half of September. In Gdańsk, the whole month of August was very
dry. Rain fell there in early September, but in the following days the drought returned and did
not stop until mid-September. The autumn was very cold. The end of the drought was not seen
in Toruń until mid-October, but complaints about the very low water level in the San River were
still being reported (Reyger, Brauer).
**5 Discussion**
Every climate proxy has its own advantages, but also its weaknesses. Therefore, to increase the
probability of correctly dating drought in Poland, we decided to use both documentary evidence
and dendrochronological data for the period before the 19th century. A satisfactory number of data
obtained from both kinds of proxies is available for period 1451–1800, allowing for reliable cross-
checking of information about the occurrence and characteristics of droughts. For the most recent
period (1801–2015), the usefulness of tree-ring data in describing dry spells (droughts) was
checked by comparing it against droughts delimited for the area of Poland using SPI calculated for
eight long-term series of monthly precipitation totals.
Tree rings in Poland can be a source of information about both hydroclimate phenomena,
such as droughts, and air temperature (Büntgen et al., 2007, 2011; Koprowski et al., 2012; Opała
and Mendecki, 2014; Opała, 2015; Pritzkow et al., 2016; Balanzategui et al., 2017). The key issue
is to isolate which factor strongly influences tree-ring growth. Up till now, tree-ring widths in
Poland have been used only for air temperature reconstructions (e.g. Przybylak et al., 2005;
Szychowska-Krąpiec, 2010; Niedźwiedź et al., 2015). In the present paper, this kind of proxy data
is used for the first time to identified droughts occurring in the vegetation period. It was assumed
that the combined information from historical and instrumental sources on the one hand, and
dendrochronological sources on the other, would be crucial in identifying the strength of water
shortage and the occurrence of droughts in Poland in recent centuries.
Extreme and severe drought occurrence in spring and summer, as identified by
documentary data, corresponds closely with the occurrence of negative pointer years (droughts).
In the period 1451–1800, 48 severe and extreme droughts have been determined to have occurred
across all of Poland or in at least two geographical regions (see Fig. 1). Dendrochronological data



showed significantly smaller rings having formed during 52.1% of these. Dobrovolný et al. (2015)
found very similar results for the Czech Republic based on a set of 3,194 oak-ring-width samples
for the last 1,250 years (761–2010). Negative tree-ring-width extremes were confirmed in
documentary sources in 53% of cases. Analysis of extreme and severe droughts that occurred in
only one geographical region in Poland reveals a better correspondence between analysed proxies
than those described earlier for the greater area of Poland (at least two regions). In this case
negative pointer years in tree rings were noted in as many as 59.1% of detected droughts by
historical sources.

9       Even better agreement between both kinds of proxy data was found when megadroughts

identified by documentary evidence were taken into account. In four (1473, 1540, 1590 and 1748)
of the six described here (see Section 4), clear signals in dendrochronological data were detected
(negative pointer years). Using documentary sources, two megadroughts (1540 and 1590) were
also qualified as very outstanding droughts in the Czech Republic (Brázdil et al., 2013). Of those,
however, only the year 1590 had a negative tree-ring width index (TRW) (of -1.818), although this
value was not very high (see Table S1 in Supplement in Dobrovolný et al., 2015). Brázdil et al.
(2013) also distinguished three other outstanding droughts in the Czech Lands (1616, 1718 and
1719). All of those also occurred in Poland, but their category using documentary evidence was
estimated by us as -2 (severe). In all those years except 1718, negative pointer years were also
found in one Polish region (see Fig. 6), while in the Czech Republic an extreme negative TRW
index (-2.474) was found only for the year 1616 (see Table S1 in Supplement in Dobrovolný et
al., 2015). Based on the published list of TRW indices for Czech Republic (oak chronology) by
Dobrovolný et al. (2015) we found 33 extreme negative TRW indices in the period 1451–1800,
which suggests favourable conditions for drought occurrence. We excluded the two last years
(1790 and 1800), which were identified for Scots pine tree rings from Upper Silesia (Opala and
Mendecki, 2014). For almost half of this set of years (48.5%), we confirmed the existence of strong
negative pointer years in Poland's tree dendrochronologies. Significantly better agreement (89%),
between the occurrence of narrow rings in the Czech Republic on the one side, and Upper Silesia
(Opała and Mendecki, 2014) and southern Poland (Opała, 2015) on the other, was found by
Dobrovolný et al. (2015) for the overlapping period 1770–1932. These quite good correspondence
patterns between negative TRW in the Czech Republic and Poland (in particular its southern part),
which are also very clear in analysis of drought occurrence and areal coverage (which are presented
in the Old World Drought Atlas [OWDA, Cook et al., 2015]), are the result of large positive sea-
level-pressure anomalies over the whole of central Europe (including Poland) in MAM and JJA
during the occurrence of negative extremes in TRW (see Fig. 5 in Dobrovolný et al., 2015).
Significantly weaker agreement (about 30%) was found between the timings of droughts in Poland
delimited using documentary evidence and droughts reconstructed for the whole of Europe using



tree rings (Cook et al., 2015). This is caused by the fact that Cook et al. (2015) used significantly
fewer dendrochronologies from Poland (only four – and those mainly from northern Poland, see
their Supplementary Materials) than we used in the present paper (22, see Table 1 for details).

4       The megadrought year of 1473 was detected in the Baltic Province on the basis of an oak

chronology from Eastern Pomerania (Ważny, 1990). Narrow rings were observed in 80 percent of
the samples for this year. The effect of the drought in 1473 can also be shifted and observed in
southern Poland in 1474 (Szychowska-Krąpiec, 2010). The drought in 1540 was observed in
different parts of Europe; particularly strong evidence is available in documentary sources (Wetter
et al., 2014; Pfister et al., 2015; Brázdil et al., 2016). Additionally, many dendrochronological data
confirm the existence of strong droughts in much of Europe, in particular from France to Latvia,
Belarus and Ukraine and from the southern Scandinavian Peninsula to northern parts of Italy
(OWDA, Cook et al., 2015). Čufar et al. (2008) identified the existence of droughts in Slovenia in
1540 based on tree rings. The scale and intensity of the 1540 megadrought in Europe described by
Wetter et al. (2014) as "an unprecedented 11-month-long Megadrought" (more severe than the
2003 drought in Western Europe and the 2010 drought in Russia) was, however, recently
questioned by Büntgen et al. (2015), who analysed this year in light of 24,303 individual tree-ring-
width measurement series. It is also worth adding here that in different parts of Europe the effect
in tree rings was shifted and observed in 1541 (Büntgen et al., 2011). Analysis of our 22
dendrochronologies reveals the occurrence of narrow rings in trees growing in the Baltic Province
and in the Lesser Poland Province, and thus not in the whole of Poland as shown in the OWDA
(Cook et al., 2015). In 1590, narrow rings were observed in the Baltic Province, but the decidedly
strongest droughts in Europe in view of this proxy were those occurring in France and Germany
(Cook et al., 2015). Narrow rings were also noted in most sites in central and eastern Europe, as
well as in Scandinavia. The year 1748 seems to have a somewhat regional character; narrow rings
were noted in the Greater Poland and Pomerania Province and in the Lesser Poland Province.
There is no information about tree reaction for this drought in selected sites in central Europe
(Büntgen et al., 2011). Looking at OWDA we see the occurrence of droughts in this year mainly
in northern and western parts of Poland (although their severity is not so large). Evidently more
severe droughts in this year in Europe were particularly observed in southern Germany, the whole
of Austria and the western borders of the Czech Republic (Cook et al., 2015).

31       Both documentary evidence and dendrochronological data clearly indicate that in the

period 1451–1800 the greatest frequency of droughts in Poland occurred in the 18[th] century, and
particularly the second half (32 cases). Similar results are also seen in the Czech Republic (see
Fig. 4a in Brázdil et al., 2013). The smallest number of droughts was noted in the 16[th] century
(about 35), and was different than in the Czech Lands, where the evidently smallest number
occurred in the 17[th] century. In the study period, the total number of all-category droughts in



Poland identified reached 148 and 156 – using documentary evidence and dendrochronological
data, respectively. This means that both proxies reconstruct quite a similar frequency of drought
occurrence in time scales from centuries to decades. The overall numbers of droughts identified
using documentary evidence in Poland (present study) and the Czech Lands (Fig. 4a in Brázdil et
al., 2013) in the overlapping period 1501–1800 were very similar and reached 132 and 126 cases,
respectively.

7       All the dendrochronologies and long-term series of precipitation that we gathered and used

for SPI calculation are available only for the common period 1876–1985. Therefore, for this
period, statistics were calculated to compare the timings of dry periods (droughts) in Poland
identified using both of these kinds of data. The agreement between droughts occurring at least in
two geographical (SPI1$_{May–Jul}$ delimited droughts) and two natural-forest regions (significant
negative pointer years) was 25.5%. On the other hand, for a less strict criterion, i.e. the occurrence
of droughts at least in one region, the agreement reached 50.9%. Thus, the latter number is close
to the value of agreement of drought timings identified using documentary evidence and the
occurrence of negative pointer years (59.1%).

16       Having those series for the abovementioned period, we also conducted a correlation

analysis to investigate how spatially coherent the association is between climate (SPI1$_{May–Jul}$) and
tree-ring widths in the area of Poland. Coefficients of Pearson's linear correlation were calculated
for 1–2 dendrochronologies representing each natural-forest region, with SPI1$_{May–Jul}$ values
calculated for long-term series of precipitation taken from meteorological stations in the same
region and closest to the area covered by the dendrochronologies. The closest relationships
between climate and tree-ring growth were obtained for the Greater Poland and Pomerania
Province and the Silesia Province, where the correlation coefficient $r$ reached: 0.40 (site 9, Poznań
in Table 1), 0.44 (site 11, Kuyavia-Pomerania) and 0.46 (site 17, Wrocław). Such good correlation
(r=0.43) was also found by Dobrovolný et al. (2015) for the Czech Republic between 18 variants
of Czech oak chronology and March–June precipitation totals. In three other Polish provinces
(Baltic Coast, Masuria and Masovia, see Fig. 1) correlation coefficients are still statistically
significant, but are clearly smaller: 0.25 (site 3, Wolin in Table 1), 0.14 (site 1, Koszalin), 0.24
(site 7, Suwałki), 0.13 (site 8, Hajnówka) and 0.21 (site 15, Warszawa). A similar correlation value
(about 0.20) between tree-ring width and precipitation in June and July was found by Helama et
al. (2014) for south-western Finland. On the other hand, a significantly better correlation (about
0.4) was calculated by Seftigen et al. (2013) for south-eastern Sweden. The increased strength of
correlation here was probably due to the selection of trees growing at xeric sites, where the radial
growth was most likely limited by moisture availability. The climate–tree-ring-growth relationship
in Lesser Poland Province was not stated – the coefficient of correlation was equal to 0.0. The



reasons for this different climate–tree-growth behaviour in this part of Poland in comparison to
other studied regions are unknown.
From the perspective of available historical sources from the period 1451–1800, an
increasing number of droughts was reported from the second half of the 16[th] century onwards,
excluding the first half of the 17[th] century. The decrease in their occurrence in this period can be
explained by large source deficiencies. They resulted from the destruction of many documents
during the the Swedish invasion on Polish territories in 1655-1660. The number of droughts in the
first half of the 17[th] century is likely to have been higher. As information about moderate droughts
is quite accidental, the sources certifying extreme and severe droughts seem more reliable and
complete. According to our research, droughts occurred most frequently in the second half of the
18[th] century. This rectifies the previously accepted data on drought in Poland. In geographic works,
it was established that in the 14[th] century there were 20 droughts in Poland, 25 in the 15[th] century,
19 in the 16[th] century, 24 in the 17[th] century, and 22 in the 18[th] century. However, the data refers
to the frequency of hot summer seasons (Sadowski 1991, Sadowski also assumed the year 1300,
1400, etc. to be the first year of a century). In later geographic works, Sadowski's findings were
accepted, assuming, however, that the hot summer seasons he had identified were characterised
by the occurrence of drought. Therefore, many subsequent Polish works mention only the number
of droughts (Słota et al., 1992; Kaca et al., 1993; Łabędzki, 2006). On the basis of the research
presented above, we conclude that droughts of greater importance (indexes -2, -3) were in fact
slightly less frequent, while their occurrence in the period from the 15[th] to the 18[th] century, as
previously stated, was slightly increasing.
6.  Summary and concluding remarks
The main results of the present paper can be summarised as follows:
1. More than one hundred droughts were found in documentary sources from the mid-15[th]
century to the end of the 18[th] century, including 17 megadroughts. A greater-than-average
number of droughts was observed in the second halves of the 17[th] and, particularly, the 18[th]
century. Dendrochronological data confirmed this general tendency in the mentioned
period. The clearly greatest number of negative pointer years occurred in the 18[th] century
and then in the period 1451–1500.
2. Droughts in the period 1451–1800 occurred most frequently in the Baltic Coast–Pomerania
and Silesia regions, while in the rest of the analysed regions their frequency was more-or-
less similar. Generally similar results have been found for the period 1722–2015 based on
instrumental data.



3. Analysis of SPI (including its lowest values – droughts) showed that the long-term frequency of droughts in Poland has been stable in the last two or three centuries.

4. Most droughts in the period 1722–2015 lasted for two months (about 60–70%), and the next most common duration was 3–4 months (10–20%). Frequencies of droughts of 5 or more months were below 10%. The longest droughts lasted for 7–8 months.

5. The frequency of all-category droughts in Poland in the period 1722–2015 was greatest in winter. his fact should be taken into account when droughts delimited using documentary evidence are analysed. In light of this information, droughts in spring and summer clearly dominated in Poland in the period 1451–1800, while in winter only three cases were mentioned.

6. Analysis of the occurrence of negative pointer years (a good proxy for droughts) showed a good correspondence with droughts delimited based on documentary and instrumental data in the periods 1451–1800 and 1722–2015, respectively.

Our study supports the usefulness of both kinds of proxy data as reliable tools for delimiting and characterising droughts for the pre-instrumental period in Poland. Information about droughts received from historical and dendrochronological data very often complete each other. In some cases where it is difficult to reliably categorise droughts based on historical sources, the occurrence of narrow rings in trees from different regions and their magnitude can significantly help in final and more reliable categorisation of this phenomenon. Such a possibility appears to be very important due to the fact that the historical data are based on subjective observations. On the other hand, the information received from old historical documents can be also useful for indicating reasons for the occurrence of the narrow rings noted in trees (droughts, vermin, etc.). As long as historical buildings in Poland continue not to be extensively investigated for wood dating, and not all historical documents are analysed for the study of old weather conditions, the knowledge about droughts will be incomplete, and futher work is thus needed.

**Competing interests.** The authors declare that they have no conflict of interest.

**Acknowledgements**. The research work of P. Oliński and R. Przybylak was supported by a grant entitled '*Climatic conditions in South Baltic Areas in the second half of the 15th and 16th centuries and their consequences for social, economic and cultural life*', funded by the National Science Centre, Poland (Grant No. DEC-2013/11/b/HS3/01458). R. Puchałka was supported by a grant entitled '*Xylogenesis and tree-ring chronologies in European beech (Fagus sylvatica) and Sessile oak (Quercus petrea) in the north-eastern margin of their natural range*', funded by the National Science Centre, Poland (Grant No. 2017/01/X/NZ8/00257).



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

Diaryusz Życia JWJmci Pana Jana Antoniego Chrapowickiego Wojewody Witebskiego [...] przekopiowany w roku
1786, T. 7 [1683-], Muzeum Narodowe w Krakowie, rps. MNKr. 169
Korespondencja Michała Dorengowskiego, Archiwum Główne Akt Dawnych, Archiwum Radziwiłłowskie, Dz. V,
3207/II
Richtsteig Johann, Chronikalische Aufzeichnung Thorner und Thorn angehende Begebenheiten, 2. vol., Archiwum
Państwowe w Toruniu, Akta miasta Torunia, Kat. II, XIII-80, XIII-80a
Żegota Pauli, Notaty z Kalendarzy dawnych krakowskich w. XV, XVI i XVII, Biblioteka Jagiellońska w Krakowie,
Rękopis nr 5358





Old prints:
Curicke Reinhold, Der Stadt Dantzig historische Beschreibung, Amsterdam und Dantzig 1688
Gomolcke Daniel, Historische Beschreibung derer grossen Theuerungen, Hunger und Kummerjahre, welche nicht
allein die k.u.k. Stadt Breslau, sondern auch dan gantze Land Schlesien bis auf das 1737 Jahr betroffen, Breslau 1737.
Kochowski Wespazjan, Annalium Poloniae ab obitu Vladislai IV. Climacter I-III, Cracoviae 1683-1698
Kronika Marcina Bielskiego (Zbiór dziejopisów polskich, 1), Warszawa 1764
Miechowita Maciej, Chronica Polonorum, Cracoviae 1521
Nicolai Henelius ab Hennenfeld Annales Silesiae, in: Silesiacarum rerum scriptores, ed. by F.W.von Sommersberg,
Leipzig 1730
Theatrum Europaeum... das ist ausführliche Beschreibung der denckwürdigsten Geschichten, so sich hin und wieder
durch Europa, wie auch in denen übrigen Welt-Theilen, von Jahr 1618 an ..., Franckfurt am Mayn 1663-1707
Zerneke J. H., Historiae Thoruniensis naufragae tabulae, Thorn 1711
The old press:
Kuryer Polski, Anno 1730 (Nr 15), 1739 (Nr 129, 133),
Kuryer Warszawski, 1763 (nr 75)
Thornische wöchentliche Nachrichten und Anzeigen nebst einem Anhange von gelehrten Sachen, Erstes Jahr 1760
Printed sources:
Annales Glogovienses bis z. J. 1493, ed. by Hermann Markgraf, in: Scriptores rerum Silesiacarum, vol. 10, Breslau
1877, p. 1-66.
Archivum vetus et novum Ecclesiae Archipresbyterialis Heilsbergensis, in: "Scriptores Rerum Warmiensium oder
Quellenschriften zur Geschichte Ermlands", ed. by Woelky Carl P., vol. 2, Braunsberg 1880, p. 587-758
Biblioteka starożytna pisarzy polskich, ed. K. Wł.. Wójcicki, Vol. 6, Warszawa 1844
Catalogus abbatum Saganensium, in: Scriptores rerum Silesiacarum, vol. 1, ed. by Gustav Adolf Stenzel, Breslau
1835, p. 173-528
Chrapowicki Jan Antoni, Diariusz, Vol 1-2: ed. by T. Wasilewski, Warszawa 1988
Chronik Michael Steinbergs, ed. by T. Schöborn, in: Scriptores rerum Silesiacarum, vol. 11 (Schweidnitzer Chronisten
des XVI. Jahrhunderts), ed. by Schimmelpfennig, Breslau 1878, p. 117-176
Die Thammendorf'sche Familienchronik, ed. by A. Schimmelpfennig, in: Scriptores rerum Silesiacarum, vol. 11
(Schweidnitzer Chronisten des XVI. Jahrhunderts), ed. by Schimmelpfennig, Schönborn, Breslau 1878, p. 1-58.
Długosza Jana Roczniki czyli Kroniki sławnego Królestwa Polskiego, 2. vol.: 12, 1445-1461 and 1462-1480, ed. by
Krzysztof Baczkowski et al., Warszawa 2009
Dupont Philippe, Pamiętniki historyi życia i czynów Jana III Sobieskiego, ed. by Dariusz Milewski, transl. by Beata
Spieralska, Warszawa 2011
Kronika walichnowska (1703-1725), wyd. ks. A. Mańkowski, Zapiski TNT, t. 8, R. 1930



Kuczyński Wiktor, Pamiętnik 1668-1737, ed. by Józef Maroszek et al., Białystok 1999
Latopisiec albo kroniczka domowa Joachima Jerlicza, ed. K. W. Wójcicki, Vol. 2, Warszawa 1853
Marcin Murinius, Kronika albo krótkie z kronik rozmaitych zebranie spraw potocznych ziemie z dawna sławney
pruskiey, ed. by K. W. Wójcik, (Biblioteka starożytnych pisarzy polskich, IV), Warszawa 1843
Nicolaus Pol, Jahrbücher der Stadt Breslau, ed. by Johann Gustav Gottlieb Büsching, Johann Gottlieb Kunisch, vol.
1-5, Breslau 1813-1824
Niezabitowski Stanisław, Dzienniki 1695-1700, oprac. i wstęp A. Sajkowski, Poznań 1998
Ossoliński Zbigniew, Pamiętnik, ed. by Józef Długosz, Warszawa 1983
Pamiętniki Łosia obejmujące wydarzenia 1646-1667, ed. by Pauli Żegota, Kraków 1858
Pasek Jan Chryzostom, Pamiętniki, ed. by Roman Pollak, Warszawa 1963
Peter Eschenloer, Geschichte der Stadt Breslau, Vol. 1-2, ed. by Gunhild Roth, Münster-NewYork 2003
Peter Eschenloer, Historia Wratislaviensis et que post mortem regis Ladislai sub electo Georgio de Podiebrat
Bohemorum rege illi acciderant prospera et adversa, ed. by Hermann Markgraf, in: Scriptores rerum Silesiacarum,
vol. 7, Breslau 1872,
Preussische Chronik des Johannes Freiberg: aus den auf der Königsberger Stadtbibliothek befindlichen Handschriften
ed. by Friedrich Adolf Meckelburg, Königsberg 1848
Radziwiłł Albrycht Stanisław, Pamiętnik o dziejach w Polsce, tom 1-3 1632-1656, oprac. A. Przyboś, R. Żelewski,
Warszawa 1980
Rocznik Jana z Targowiska 1386-1491, ed. by Emil Kalitowski, in: Monumenta Poloniae Historica, vol. 3, Lwów
1878, p. 232-240
Rocznik wrocławski dawny 1238-1308 i rocznik magistratu wrocławskiego 1149-1491, ed. by August Bielowski, in:
Monumenta Poloniae Historica, vol. 3, Lwów 1878, p. 680-688
Rzączyński Gabriel, Historia Naturalis Curiosa Regni Polniae, Magni Ducatus Lituaniae, annexarumque
Provinciarum in tractatus XX divisa, Sandomiriae 1721
Sigismundi Rosiczii Chronica et numerus episcoporum Wratislaviensium, in: Scriptores rerum Silesiacarum, ed. by
F. Wächter, Breslau 1883, vol. 12, p. 29-86
Skoraczewski F., Materyały do historyi Miłosławia, Poznań 1910
Spominki kazimierskie 1422-1473, ed. by August Bielowski, in: Monumenta Poloniae Historica, vol. 3, Lwów 1878,
p. 242-243.
Świętosława Orzelskiego Bezkrólewia ksiąg ośmioro 1572-1576, ed. by Edward Kuntze, in: Scriptores rerum
Polonicarum, vol. 22, Kraków 1917
Wernicke Julian Emil, Geschichte Thorns aus Urkunden, Dokumenten und Handschriften, 2. vol., Thorn 1839-1842