# Peer review of "Droughts in the area of Poland in recent centuries in the light of multiproxy data"

_Climate of the Past, 2019_

## Referee Comment (RC1) · Anonymous Referee #1 · 10 Jul 2019

Dear authors, I highly appreciate the approach of combining and complementing available tree-ring width chronologies, written documentary accounts and instrumental data to investigate droughts, i.e. their occurrence, frequency and intensity, in Poland back to ca. 900 CE. An amazing 200 documented drought accounts for the period 1451–1800 were collected and categorized into three classes of severity. In addition, 22 tree-ring width chronologies were used to detect years of extreme low annual growth, so-called negative pointer years, which were attributed to drought events. The extension into the industrial period was done using the Standardized Precipitation Index (SPI) with different seasonal lengths and which was calculated on eight long precipitation records.

Overall, the comprehensive analysis of drought events and duration using existing proxy data is needed, especially under current climate change. However, the amount

of statistical approaches applied make the study partly difficult to understand. Moreover, there are several shortcomings in the manuscript regarding the structure and content. Substantial improvements should be made prior to publication and I strongly recommend that the English be revised by a professional service or a native English-speaking scientist working in the field.

General comments

- The title is not reflecting the study very well, maybe include that a multi-proxy approach was used or highlight the main result, for example.

- The abstract needs shortening and a clear structure by including a motivation of the study which is followed by data, methods, results and conclusion/significance of the study. The abstract should not be too long and should not include references.

- The introduction needs improvement by 1) removing unnecessary information e.g. reduce p.3, l. 17-20, 2) write in a more precise way e.g. p.2, l.10. "statistical analyses" of what?, and 3) provide more information e.g. p.3, l.21. in which areas is drought the most stressful factor – to only provide a few examples. Also, I was wondering why the authors cite four lines of a publication on l. 18-21? This can be summarized.

- Structure of the Data and Methods chapters needs improvement. A straightforward description of the documentary data is missing. After reading the chapter 2.1, it is not entirely clear what data from whom were used. Maybe start with the summarizing paragraph (p.5, l. 30 – p.6., l.18) and add some (and only) important information from the paragraphs before. For the dendrochronological data, no information about the quality of the individual tree-ring width chronologies is provided. Information of the number of samples, inter-series correlation, mean segment lengths can be easily added in Table. 1. Information on the sample replication in a tree-ring width chronology is essential to evaluate drought events that were detected during a low replicated time period.

For the Method chapter, the examples of the individual drought classes in chapter

3.1 are quite long. Please, consider reduction to only 2 to 3 examples and place the remaining examples in the supplementary material.

On page 19, chapter "2.3 Instrumental data" needs to be moved into "2. Data chapter". Instead there should be a clearly written paragraph about the detection of the climate-growth relationships of all tree-ring width chronologies, for which period and for what climate variables. Why not use the SPI data for the analysis of the climate response of the trees which would simplify the entire study a lot and at the same time, prove your hypotheses (p.18, l.9)?

- Description of the methods lacks detailed and important information. For example, on p. 18, l. 14 "climate monthly precipitation and temperature" were used to evaluate the climate growth relationship. However, only results for precipitation are shown in Fig. 2 and information of the period over which the correlation was done is missing.

- Methodology for the evaluation of the climate-growth relationship is not sufficient. Firstly, it is not clear if the age trend from the individual tree-ring width series is removed and what method was applied. Secondly, it is questionable if daily precipitation data need to be used given 1) that this led the authors to a generalization which might be not true (p.19, l.4) and 2) the description and mention of the droughts in the documentary data are not on daily resolution either. Moreover, I would like to see a comprehensive climate-growth analysis of all tree-ring width chronologies with information of species, Pearson correlation coefficients, period of correlation etc., at least in a Table. This is very important since a publication by Przybylak et al. 2005 used a tree-ring width chronology from pine (Pinus sylvestris) to reconstruct mean January – April air temperature for Poland.

- Please avoid repetitions, e.g., on p.19, l.7-11: the two sentences are the same.

- P.19, l.11-16: please rephrase and clarify this entire paragraph since it is not clear what was done and why.

---

## Referee Comment (RC2) · Anonymous Referee #2 · 11 Jul 2019

This article contains a useful review, and assessment, of the occurrence and severity of drought in Poland during the past five centuries using both instrumental data (from the 18th century), historical documentary data and tree-ring width data. As past drought, or hydroclimate in general, in Poland is an under-researched topic, the manuscript is clearly worth publication after revision. The manuscript is in need of some polishing and English language editing but can otherwise, in my opinion, be published. That said, I would still recommend the authors to consider a few things:

1) Streamline part of the text, including the Abstract and the Introduction, as especially the Abstract is too long and too detailed. Moreover, part of the Introduction does not really well capture the state-of-the-art knowledge of hydroclimatic changes with global warming and the selection of references in the introduction is a bit biased.

2) The translation of narrow tree-rings to dry years/growing seasons are a bit problematic as the response between tree-growth and hydroclimate is non-linear, and not stable over time, and low temperatures may also produce narrow rings. My concern here is mainly that some of the narrow rings during the climax of the Little Ice Age c. 1570–1710, as well as during some other shorter time intervals, may in some cases be a result of very cold springs and summers. The authors could probably systematically compare the narrow rings with climate information in the documentary sources to rule this possibility out. It is a bit unclear in the present version of the manuscript if this has been done or not. Regarding the non-linear relationship between tree growth and climate, see the discussion and references given in: https://iopscience.iop.org/article/10.1088/1748-9326/ab2c7e

3) I would recommend the authors to better include, and cite, the recent scholarship in historical climatology. A good starting point, with ample references, could be the articles in The Palgrave Handbook of Climate History, ed. S White et al (London: Palgrave Macmillan).

Minor comments:

Page 2 (in general): The evidence for increasing droughts in recent decades is weaker, and more controversial, than evident from what the authors write. To a large extent, the results are dependent on which drought metrics is used. It is also questionable, except in some particular regions, if there is any empirical evidence for longer breaks between episodes of precipitation. The present reviewer has in the past six years worked considerably with hydroclimate and not found support for this in the literature.

Page 2, line 5: "The increase in degree of" is a strange formulation here.

Page 2, line 16: Cite also: Greve P et al 2014 Global assessment of trends in wetting and drying over land Nat. Geosci. 7 716–21

Page 3, lines 30–33: I guess the authors provide these examples to show that hydrocli-

mate reconstructions also can be obtained for rather cold regions of Europe? It should be made clearer here.

Page 12, lines 12–14: It should be better pointed out that some statements about rivers that had dried out certain summers likely are not reliable or that they, at least, are overstatements.

Section 2.2 and section 3.2: This must be placed in a better dendro research context. In particular, the non-linear relationships between temperature and hydroclimate and tree-growth need to be discussed. I also note that the correlation between the tree-ring records and precipitation is very weak. It is far weaker than in tree-ring chronologies explicitly developed for reconstructing hydroclimate. I think it is important, and fair to the reader, to point out that many of the included tree-ring chronologies have not be developed with that purpose explicitly in mind.

Page 23, lines 7–10: This part can be shortened as it is not very clear what is meant with that drought has not been "very frequent".

Page 25, lines 15–16: This is an interesting and potentially important part. Could it also be that there were fewer droughts in the first half of the 17th century in Poland because it also was the coldest part of the Little Ice Age with less evapotranspiration due to lower temperatures?

Page 26, line 31: Very strange formulation. Please, consider revision.

Section 3.2 and Fig. 7: The low number of dry pointer years in the medieval times is certainly a result of fewer records. This should be pointed out as dry years in the region actually seem to have been more frequent back in medieval times. See, most recently: Scharnweber, T., Heußner, K.-U., Smiljanic, M., Heinrich, I., van der Maaten-Theunissen, M., van der Maaten, E., Struwe, T., Buras, A., Wilmking, M., 2019. Removing the no-analogue bias in modern accelerated tree growth leads to stronger medieval drought. Sci. Rep. 9, 2509. https://doi.org/10.1038/s41598-019-39040-5.

Page 33: Try to make changes in drought trends over time clearer to the reader. As it is written now, it is a bit hard to follow this.

Page 44, lines 18–21: The formulation is unclear and a bit hard to follow.

Page 45, line 7: "T" is missing in "This".

Page 45, line 22: Insects rather than vermin.
* * *

---

## Author Comment (AC1) · 2 Sep 2019

Dear Editor,

Thank you very much for giving us the opportunity to respond to the reviewers' remarks, suggestions, etc. We are very grateful for all their opinions and suggestions, which were usually very helpful and constructive. All passages where changes in the article have been made are also placed in the text below.

All changes in the text are marked in red fonts. For clarity the reviewers' texts are in black and our answers are in blue. Corrected attached passages are shown in italic.

We would kindly like to inform you that the following alterations have been made to the text regarding the reviewers' comments, suggestions and opinions:

**Anonymous Referee #1**

Dear authors, I highly appreciate the approach of combining and complementing available tree-ring width chronologies, written documentary accounts and instrumental data to investigate droughts, i.e. their occurrence, frequency and intensity, in Poland back to ca. 900 CE. An amazing 200 documented drought accounts for the period 1451–1800 were collected and categorized into three classes of severity. In addition, 22 tree-ring width chronologies were used to detect years of extreme low annual growth, so-called negative pointer years, which were attributed to drought events. The extension into the industrial period was done using the Standardized Precipitation Index (SPI) with different seasonal lengths and which was calculated on eight long precipitation records.

Overall, the comprehensive analysis of drought events and duration using existing proxy data is needed, especially under current climate change. However, the amount of statistical approaches applied make the study partly difficult to understand. Moreover, there are several shortcomings in the manuscript regarding the structure and content. Substantial improvements should be made prior to publication and I strongly recommend that the English be revised by a professional service or a native Englishspeaking scientist working in the field.

The entire text was corrected by a native speaker.

General comments

- The title is not reflecting the study very well, maybe include that a multi-proxy approach was used or highlight the main result, for example.

Answer: was changed according to the reviewer's suggestion.

Final title: *Droughts in the area of Poland in recent centuries in the light of multi-proxy data*

- The abstract needs shortening and a clear structure by including a motivation of the study which is followed by data, methods, results and conclusion/significance of the study. The abstract should not be too long and should not include references.

Answer: was shortened and all remarks of the Reviewer were taken into account

**Final text of Abstract**: *The history of drought occurrence in Poland in the last millennium is poorly known. To improve this knowledge we have conducted a comprehensive analysis using both proxy data (documentary and*

*dendrochronological) and instrumental measurements of precipitation. The paper presents the main features of droughts in Poland in recent centuries, including their frequency of occurrence, coverage, duration and intensity. The reconstructions of droughts based on all the mentioned sources of data covered the period 996–2015. Examples of megadroughts were also chosen using documentary evidence, and some of them were described.*

*Various documentary sources have been used to identify droughts in the area of Poland in period 1451–1800 and to estimate their intensity, spatial coverage and duration. Twenty-two local chronologies of trees (pine, oak, and fir) from Poland were taken into account for detecting negative pointer years (exceptionally narrow rings). The longest chronology covers the years 996–1986 and was constructed for eastern Pomerania. The delimitation of droughts based on instrumental data (eight long-term precipitation series) was conducted using two independent approaches (Standard Precipitation Index (SPI) calculated for 1-, 3-, and 24-month time scales, and new method proposed by authors). For delimitation of droughts (dry months), the criteria used were those proposed by McKee and modified for the climate conditions of Poland by Łabędzki.*

*More than one hundred droughts were found in documentary sources in the period 1451–1800, including 17 megadroughts. A greater-than-average number of droughts was observed in the second halves of the 17th century, and of the 18th century in particular. Dendrochronological data confirmed this general tendency in the mentioned period. The clearly greatest number of negative pointer years occurred in the 18th century and then in the period 1451–1500. In the period 996–2015, a total of 758 negative pointer years were recorded.*

*Analysis of SPI (including its lowest values, i.e. droughts) showed that the long-term frequency of droughts in Poland has been stable in the last two or three centuries. Extreme and severe droughts were most frequent in the coastal part of Poland and in Silesia. Most droughts had a duration of two months (about 60–70%), or 3–4 months (10–20%). Frequencies of droughts with a duration of 5 and more months were lower than 10%. The longest droughts had a duration of 7–8 months. The frequency of droughts of all categories in Poland in the instrumental period 1722–2015 was greatest in winter, while the documentary evidence (1451–1800) rarely mentions droughts in this season. The occurrence of negative pointer years (a good proxy for droughts) was compared with droughts delimited based on documentary and instrumental data. A good correspondence was found between the timing of occurrence of droughts identified using all three kinds of data (sources).*

- The introduction needs improvement by 1) removing unnecessary information e.g. reduce p.3, l. 17-20, 2) write in a more precise way e.g. p.2, l.10. "statistical analyses" of what?,

Answer: The Introduction part was corrected and the present version is as below:

[revised manuscript text omitted]

and 3) provide more information e.g. p.3, l.21. in which areas is drought the most stressful factor – to only provide a few examples.

Answer: We updated the text with information about areas with the most frequent occurrence of drought in Poland. The following passage was added:

*According to studies by Somorowska (2016) the effect of drought extends from the south-west towards the centre of the country and, in some cases, to the north-east of Poland. Another study suggest that in the future some of the highest probabilities of drought occurrence may be in the central part, with the lowest probability in south-eastern Poland (Diakowska et al., 2018).*

Also, I was wondering why the authors cite four lines of a publication on l. 18-21? This can be summarized.

Answer: Sorry, but in the mentioned lines there is no citing of publications in any of page in the entire manuscript? It must be a mistake?

- Structure of the Data and Methods chapters needs improvement. A straightforward description of the documentary data is missing. After reading the chapter 2.1, it is not entirely clear what data from whom were used. Maybe start with the summarizing paragraph (p.5, l. 30 – p.6., l.18) and add some (and only) important information from the paragraphs before.

Answer: It seems to us that it is precisely written in the text from which historical sources weather excerpts were taken. The structure of the chapter 2.1 is typical from the historical point of view and therefore we have decided to leave the chapter as it is. At the beginning the published sources are described, divided into different types, and then the archival sources (not published). At the end of the chapter a short summing-up is given. It is rather difficult and in our opinion not appropriate to summarize this not-too-long chapter.

For the dendrochronological data, no information about the quality of the individual tree-ring width chronologies is provided. Information of the number of samples, inter-series correlation, mean segment lengths can be easily added in Table. 1. Information on the sample replication in a tree-ring width chronology is essential to evaluate drought events that were detected during a low replicated time period.

Answer: Table 1 was updated. We provided the information about number of samples, EPS and rbar.tot. In the case of Site 12 the EPS is extremely low; however, the chronologies were not used for climate reconstruction but for detecting negative pointer years. Pointer years confirmed the information from historical sources and show that drought can also affect the trees. It is also worthwhile to note after Buras (2017) that *"EPS is a measure of how well a finite sample of tree-ring data represents an infinite population chronology, but it will not necessarily indicate whether a tree-ring chronology is suitable for climate reconstruction purposes."*

Table 1

| Site number | Site name | Time span | Number of samples | EPS | rbar.tot | Species | Source |
|---|---|---|---|---|---|---|---|
| Region I (Baltic Province) | | | | | | | |
| Site 1 | Koszalin | 1782–1987 | 22 | 0.899 | 0.339 | Oak | https://www.ncdc.noaa.gov/ (Ważny, 1990) |
| Site 2 | Gdańsk | 1762–1986 | 45 | 0.887 | 0.192 | Oak | https://www.ncdc.noaa.gov/ (Ważny, 1990) |
| Site 3 | Wolin | 1554–1987 | 23 | 0.877 | 0.318 | Oak | https://www.ncdc.noaa.gov/ (Ważny, 1990) |

| Site 4 | Gdańsk | 1175–1396 | 13 | 0.579 | 0.388 | Oak | Dąbrowski HP, unpublished |
|--------|--------|-----------|----|-------|-------|-----|---------------------------|
| Site 5 | western Pomerania | 996–1986 | 205 | 0.907 | 0.250 | Oak | https://www.ncdc.noaa.gov/ (Ważny, 1990) |
| Region II (Masuria-Podlasie Province) | | | | | | | |
| Site 6 | Gołdap | 1871–1987 | 22 | 0.941 | 0.472 | Oak | https://www.ncdc.noaa.gov/ (Ważny, 1990) |
| Site 7 | Suwałki | 1861–1987 | 19 | 0.872 | 0.303 | Oak | https://www.ncdc.noaa.gov/ (Ważny, 1990) |
| Site 8 | Hajnówka | 1720–1985 | 19 | 0.851 | 0.314 | Oak | https://www.ncdc.noaa.gov/ (Ważny, 1990) |
| Region III (Greater Poland-Pomerania Province) | | | | | | | |
| Site 9 | Poznań | 1836–1987 | 17 | 0.904 | 0.385 | Oak | https://www.ncdc.noaa.gov/ (Ważny, 1990) |
| Site 10 | Zielona Góra | 1774–1987 | 19 | 0.876 | 0.330 | Oak | https://www.ncdc.noaa.gov/ (Ważny, 1990) |
| Site 11 | Toruń | 1714–2015 | 48 | 0.886 | 0.335 | Oak | Puchałka et al., 2016 (updated) |
| Site 12 | Tuchola | 1249–1490 | 7 | 0.054 | 0.347 | Pine | Dąbrowski HP, unpublished |
| Site 13 | Kuyavia-Pomerania | 1169–2015 | 247 | 0.816 | 0.195 | Pine | Koprowski et al., 2012 |
| Site 14 | Chojnice | 1100–1468 | 21 | 0.688 | 0.327 | Oak | Dąbrowski HP, unpublished |
| Region IV (Masovia-Podlasie Province) | | | | | | | |
| Site 15 | Warszawa | 1690–1985 | 19 | 0.850 | 0.291 | Oak | https://www.ncdc.noaa.gov/ (Ważny, 1990) |
| Region V (Silesia Province) | | | | | | | |
| Site 16 | Upper Silesia | 1770–2010 | 80 | 0.880 (average) | correlation 0.530 | Pine and oak | Opała and Mendecki, 2014 |
| Site 17 | Wrocław | 1727–1987 | 22 | 0.870 | 0.327 | Oak | https://www.ncdc.noaa.gov/ (Ważny, 1990) |
| Site 18 | Upper Silesia | 1568–2010 | 178 | 0.850 | correlation 0.510 | Pine | Opała, 2015 |
| Region VI (Lesser Poland Province) | | | | | | | |
| Site 19 | Kraków | 1792–1986 | 29 | 0.906 | 0.361 | Oak | https://www.ncdc.noaa.gov/ (Ważny, 1990) |
| Site 20 | Kosobudy | 1782–1989 | 22 | 0.937 | 0.448 | Oak | https://www.ncdc.noaa.gov/ (Ważny, 1990) |
| Site 21 | Lesser Poland | 1109–2004 | 238 | No data | No data | Pine | Szychowska-Krąpiec, 2010 |
| Site 22 | Lesser Poland | 1109–2006 | 560 | No data | No data | Fir | Szychowska-Krąpiec, 2010 |

For the Method chapter, the examples of the individual drought classes in chapter 3.1 are quite long. Please, consider reduction to only 2 to 3 examples and place the remaining examples in the supplementary material.

Answer: the examples of the individual drought classes in chapter 3.1 were significantly reduced according to the reviewer's suggestion, see tables 3–5 below:

Table 3

[revised manuscript text omitted]

On page 19, chapter "2.3 Instrumental data" needs to be moved into "2. Data chapter".

Answer: No. This is just an error. The numbering of the subchapter should be 3.3 and not 2.3 as it is in the original version. We corrected the numbering.

Instead there should be a clearly written paragraph about the detection of the climategrowth relationships of all tree-ring width chronologies, for which period and for what climate variables. Why not use the SPI data for the analysis of the climate response of the trees which would simplify the entire study a lot and at the same time, prove your hypotheses (p.18, l.9)?

Answer: SPI was not taken into account because this parameter results directly from precipitation data.

- Description of the methods lacks detailed and important information. For example, on p. 18, l. 14 "climate monthly precipitation and temperature" were used to evaluate the climate growth relationship. However, only results for precipitation are shown in Fig. 2 and information of the period over which the correlation was done is missing.

Answer: additional information were added in the new Table 6. The information about temperature was also updated (see text below and Table 6).

*For each site the climate growth relationships were tested against monthly precipitation and temperature data starting from 1951 and covers maximum time span depending on the length of the chronology (Table 6). Because the time span was too short (for example for Site 2 when chronology covers the years 1951-1986) for some extended analysis going back to previous months, the common period from previous October to current September was taken into account.*

Table 6. Climate growth relationships for analysed sites. Only the highest correlation coefficients are presented – with level of significance $p < 0.05$.

| Site number | Site name | Analysed period | Highest Pearson correlation coefficient | Months with highest correlation coefficient | Meteorological station | Species | Source |
|---|---|---|---|---|---|---|---|
| Region I (Baltic Province) | | | | | | | |
| Site 1 | Koszalin | 1951–1987 | 0.378 | Sum of precipitation from May to June | Koszalin | Oak | https://www.ncdc.noaa.gov/ (Ważny, 1990) |
| Site 2 | Gdańsk | 1951–1986 | 0.296 (not significant) | Sum of precipitation from June to July | Gdańsk | Oak | https://www.ncdc.noaa.gov/ (Ważny, 1990) |
| Site 3 | Wolin | 1951–1987 | 0.565 | Sum of precipitation from June to August | Świnoujście | Oak | https://www.ncdc.noaa.gov/ (Ważny, 1990) |
| Site 4 | Gdańsk | 1175–1396 | No climate data | No climate data | No climate data | Oak | Dąbrowski HP, unpublished |
| Site 5 | western Pomerania | 1951–1986 | 0.456 | Sum of precipitation from June to July | Koszalin | Oak | https://www.ncdc.noaa.gov/ (Ważny, 1990) |
| Region II (Masuria-Podlasie Province) | | | | | | | |
| Site 6 | Gołdap | 1951–1987 | 0.589 | Temperature current May | Suwałki | Oak | https://www.ncdc.noaa.gov/ (Ważny, 1990) |
| Site 7 | Suwałki | 1951–1987 | 0.50 | Sum of precipitation from June to July | Suwałki | Oak | https://www.ncdc.noaa.gov/ (Ważny, 1990) |
| Site 8 | Hajnówka | 1951–1985 | 0.285 | Sum of precipitation from July to August | Białystok | Oak | https://www.ncdc.noaa.gov/ (Ważny, 1990) |
| Region III (Greater Poland-Pomerania Province) | | | | | | | |
| Site 9 | Poznań | 1951–1987 | 0.485 | Sum of precipitation from May to July | Poznan | Oak | https://www.ncdc.noaa.gov/ (Ważny, 1990) |
| Site 10 | Zielona Góra | 1951–1987 | -0.322 | Temperature, previous December | Gorzów Wielkopolski | Oak | https://www.ncdc.noaa.gov/ (Ważny, 1990) |
| Site 11 | Toruń | 1951–2015 | 0.334 -0.334 | Sum of precipitation from May to June, temperature in June | Toruń | Oak | Puchałka et al., 2016 (updated) |
| Site 12 | Tuchola | 1249–1490 | No climate data | No climate data | No climate data | Pine | Dąbrowski HP, unpublished |
| Site 13 | Kuyavia-Pomerania | 1951–2015 | 0.443 | Sum of precipitation from May to July | Toruń | Pine | Koprowski et al., 2012 |
| Site 14 | Chojnice | 1100–1468 | No climate data | No climate data | No climate data | Oak | Dąbrowski HP, unpublished |
| Region IV (Masovia-Podlasie Province) | | | | | | | |

| Site | Location | Period | Value | Month/Climate | City | Species | Source |
|------|----------|--------|-------|---------------|------|---------|--------|
| Site 15 | Warszawa | 1951–1985 | -0.316 | Temperature, previous December | Warszawa | Oak | https://www.ncdc.noaa.gov/ (Ważny, 1990) |
| Region V (Silesia Province) | | | | | | | |
| Site 16 | Upper Silesia | 1886–1984 | >0.4 Precipitation data not presented due to lower statistical significance | Temperature of February and March for pine | Opole, Wrocław, Katowice and Racibórz | Pine and oak | Opała and Mendecki, 2014 |
| Site 17 | Wrocław | 1951–1987 | 0.376 | Sum of precipitation from May to June, | Wrocław | Oak | https://www.ncdc.noaa.gov/ (Ważny, 1990) |
| Site 18 | Upper Silesia | 1568–2010 | Only pointer years were analysed | | | Pine | Opała, 2015 |
| Region VI (Lesser Poland Province) | | | | | | | |
| Site 19 | Kraków | 1915–1986 | 0.324 (not significant) | Temperature in February | Kraków | Oak | https://www.ncdc.noaa.gov/ (Ważny, 1990) |
| Site 20 | Kosobudy | 1951–1989 | 0.314 -0.323 | Sum of precipitation from May to July, temperature in June | Lublin and Radawiec | Oak | https://www.ncdc.noaa.gov/ (Ważny, 1990) |
| Site 21 | Lesser Poland | 1881-1999 | >0.4 | Temperature in March | Kraków | Pine | Szychowska-Krąpiec, 2010 |
| Site 22 | Lesser Poland | 1881-1999 | >0.4 | Temperature in February | Kraków | Fir | Szychowska-Krąpiec, 2010 |

- Methodology for the evaluation of the climate-growth relationship is not sufficient. Firstly, it is not clear if the age trend from the individual tree-ring width series is removed and what method was applied. Secondly, it is questionable if daily precipitation data need to be used given 1) that this led the authors to a generalization which might be not true (p.19, l.4) and 2) the description and mention of the droughts in the documentary data are not on daily resolution either. Moreover, I would like to see a comprehensive climate-growth analysis of all tree-ring width chronologies with information of species, Pearson correlation coefficients, period of correlation etc., at least in a Table. This is very important since a publication by Przybylak et al. 2005 used a tree-ring width chronology from pine (Pinus sylvestris) to reconstruct mean January – April air temperature for Poland.

Answer: we added the passage describing detrending methods used by us, see text below

*De-trending of the chronology was done with the dplR software (Bunn 2008) using the smoothing spline option, which reflects trends in the chronology better than other options. The ''n-year spline'' was fixed at two thirds the wavelength of n years (Cook et al. 1990). The residual version of the chronology was built by pre-whitening, performed by fitting an autoregressive model to the data with AIC model selection (Bunn 2008).*

Daily data shows more precisely the period of the year which influences tree growth. We used this analysis to prove assumptions about the effect of drought on trees creating narrow rings. For the rest of the comments please see the text below and Table 6.

*For each site the climate growth relationships were tested against monthly precipitation and temperature data starting from 1951 and covers maximum time span depending on the length of the chronology (Table 6). Because the time span was too short (for example for Site 2 when chronology covers the years 1951-1986) for some extended analysis going back to previous months, the common period from previous October to current September was taken into account.*

- Please avoid repetitions, e.g., on p.19, l.7-11: the two sentences are the same.

Answer: Repeated parts were deleted

- P.19, l.11-16: please rephrase and clarify this entire paragraph since it is not clear what was done and why.

Answer: Paragraph was updated to the following version:

*The optimal window of days was revealed to be from May 6 to August 3 for pine with maximal correlation coefficient 0.435, and from April 21 to July 19 for oak with maximal correlation coefficient 0.305.*

**Anonymous Referee #2**

This article contains a useful review, and assessment, of the occurrence and severity of drought in Poland during the past five centuries using both instrumental data (from the 18th century), historical documentary data and tree-ring width data. As past drought, or hydroclimate in general, in Poland is an under-researched topic, the manuscript is clearly worth publication after revision. The manuscript is in need of some polishing and English language editing but can otherwise, in my opinion, be published.

The entire text was corrected by a native speaker.

That said, I would still recommend the authors to consider a few things:

1) Streamline part of the text, including the Abstract and the Introduction, as especially the Abstract is too long and too detailed.

Answer: was corrected, the first Reviewer also gave the same remarks. For text see reply to the first Reviewer.

Moreover, part of the Introduction does not really well capture the state-of-the-art knowledge of hydroclimatic changes with global warming and the selection of references in the introduction is a bit biased.

Answer: We have introduced some changes to the *Introduction* Section according to the Reviewer's suggestion (for details, please see the text in the reply to the first Reviewer). We hope that now the Introduction presents the real state and is not biased.

2) The translation of narrow tree-rings to dry years/growing seasons are a bit problematic as the response between tree-growth and hydroclimate is non-linear, and not stable over time, and low temperatures may also produce narrow rings. My concern here is mainly that some of the narrow rings during the

climax of the Little Ice Age c. 1570–1710, as well as during some other shorter time intervals, may in some cases be a result of very cold springs and summers. The authors could probably systematically compare the narrow rings with climate information in the documentary sources to rule this possibility out. It is a bit unclear in the present version of the manuscript if this has been done or not. Regarding the non-linear relationship between tree growth and climate, see the discussion and references given in: https://iopscience.iop.org/article/10.1088/1748-9326/ab2c7e

Answer: We are aware of these limitations. Pointer years confirmed the information from historical sources and show that drought can also affect the trees.

3) I would recommend the authors to better include, and cite, the recent scholarship in historical climatology. A good starting point, with ample references, could be the articles in The Palgrave Handbook of Climate History, ed. S White et al (London: Palgrave Macmillan).

Answer: Thank you for this recommendation. According to the Reviewer's suggestion the mentioned publication, which is important for general knowledge about drought occurrence in the world and their environmental and societal consequences, was cited. There is a myriad of publications dealing with the issue of droughts, thus the authors tried to cite the most important of them. To our knowledge the most important publication items dealing with the history of drought occurrence in Poland and central Europe in the last millennium are included in the paper.

Minor comments:

Page 2 (in general): The evidence for increasing droughts in recent decades is weaker, and more controversial, than evident from what the authors write. To a large extent, the results are dependent on which drought metrics is used.

Answer: The remarks of the reviewer were taken into account, the text has been changed and we hope that we have fulfilled the reviewer's expectation, see the Introduction chapter in the reply to the first Reviewer.

It is also questionable, except in some particular regions, if there is any empirical evidence for longer breaks between episodes of precipitation.

Answer: But we wrote (see lines 5-8) that this statement concerns only "... some regions".

The present reviewer has in the past six years worked considerably with hydroclimate and not found support for this in the literature.

Answer: we added one more reference showing the small changes in drought occurrence and the sentence that the issue still needs more research. See again the Introduction chapter in the reply to the first Reviewer.

Page 2, line 5: "The increase in degree of" is a strange formulation here.

Answer: was corrected to: The increase in rate  of global warming.

Page 2, line 16: Cite also: Greve P et al 2014 Global assessment of trends in wetting and drying over land Nat. Geosci. 7 716–21

Answer: citation was added.

Page 3, lines 30–33: I guess the authors provide these examples to show that hydroclimate reconstructions also can be obtained for rather cold regions of Europe? It should be made clearer here.

Answer: The text was changed to: *Also in other countries lying near Poland, such as* Finland (Helama and Lindholm, 2003), Sweden (Seftigen et al., 2013) and Czech Republic (Dobrovolný et al., 2015) the relationships are significant enough to reconstruct drought.

Page 12, lines 12–14: It should be better pointed out that some statements about rivers that had dried out certain summers likely are not reliable or that they, at least, are overstatements.

Answer: It seems to us that the information that the reviewer suggest to include in this passage is, in reality, present in the original text, see text below:

*Sometimes, and probably in an exaggerated way, sources reported the drying up of smaller rivers.*

For this reason, this kind of information was treated by us very carefully.

Section 2.2 and section 3.2: This must be placed in a better dendro research context. In particular, the non-linear relationships between temperature and hydroclimate and tree-growth need to be discussed. I also note that the correlation between the tree-ring records and precipitation is very weak. It is far weaker than in tree-ring chronologies explicitly developed for reconstructing hydroclimate. I think it is important, and fair to the reader, to point out that many of the included tree-ring chronologies have not be developed with that purpose explicitly in mind.

Answer: After the taking into account only narrow rings and precipitation in selected period from daily data the correlation coefficient is 0.79 ($p<0.05$) for pine, and 0.65 ($p<0.05$) for oak.

Page 23, lines 7–10: This part can be shortened as it is not very clear what is meant with that drought has not been "very frequent".

Answer: the passage was shortened according to the Reviewer's suggestion, see below the present version:

*Records on drought for historical reconstruction of climate can be found in many different historical sources from Poland. Their number has significantly increased since the mid-15$^{th}$ century, which is why the mid-15$^{th}$ century was adopted as the initial chronological boundary for the reconstruction of the number and intensity of droughts in the Polish territory using documentary evidence.*

Page 25, lines 15–16: This is an interesting and potentially important part. Could it also be that there were fewer droughts in the first half of the 17th century in Poland because it also was the coldest part of the Little Ice Age with less evapotranspiration due to lower temperatures?

Answer: According to the reconstruction made by Przybylak et al. (2005) this period had the same winter and summer temperatures as the neighbouring historical periods. Therefore we rather prefer to leave the text as it is.

Page 26, line 31: Very strange formulation. Please, consider revision.

Answer: Text was changed to: *More chronologies in the last 300 years result from existing living trees.*

Section 3.2 and Fig. 7: The low number of dry pointer years in the medieval times is certainly a result of fewer records. This should be pointed out as dry years in the region actually seem to have been more frequent back in medieval times. See, most recently: Scharnweber, T., Heußner, K.-U., Smiljanic, M., Heinrich, I., van der MaatenTheunissen, M., van der Maaten, E., Struwe, T., Buras, A., Wilmking, M., 2019. Removing the no-analogue bias in modern accelerated tree growth leads to stronger medieval drought. Sci. Rep. 9, 2509. https://doi.org/10.1038/s41598-019-39040-5.

Answer: We agree with the Reviewer's opinion and therefore the following text was added:

*However the small number of pointer years from 996 to 1200 may be related to the low number of samples. This period is called the "medieval climate anomaly" and reconstruction for northern-central Europe revealed considerably drier conditions for these years (Scharnweber et al., 2019).*

Page 33: Try to make changes in drought trends over time clearer to the reader. As it is written now, it is a bit hard to follow this.

Answer: According to the Reviewer's suggestions some changes (see text below) were introduced to the text. We hope that now the passage is more clear for readers.

*In winter, extreme droughts do not show any significant changes over time, but it should be emphasised here that they were slightly more frequent in 1951–2000 than in 1851–1900. In spring, moderate droughts prevailed still in the period 1851–1950 (usually 4–6 cases), with a greater frequency in the earlier 50-year period. Both severe and extreme droughts were most frequent (usually 1–3 cases) in both 1851–1900 and, in particular, 1951–2000 (Fig. 10). In summer, there is a clear change in the time pattern of drought occurrence: drought frequency rises in the 20th century (except severe droughts), and in the case of moderate droughts particularly in its second half. The contrast in drought The frequency of extreme droughts is evidently higher in between the 20th century compared to the pre-1900 period. is very clear, primarily in the case of extreme droughts. In autumn, moderate droughts do not show great changes in the last two centuries, while severe and extreme droughts were most frequent in the first and second halves of the 20th century, respectively (Fig. 10).*

Page 44, lines 18–21: The formulation is unclear and a bit hard to follow.

Answer: the text was rewritten and its final state is:

*On the basis of the research presented in this paper, we conclude that severe and extreme droughts of greater importance (indexes -2, -3, respectively) were in fact slightly less frequent, while their occurrence was increasing slightly in the period from the 15th to the 18th century, as previously stated*

Page 45, line 7: "T" is missing in "This".

Answer: was corrected

Page 45, line 22: Insects rather than vermin.

Answer: was corrected

---

## Author Comment (AC2) · 16 Sep 2019

Dear Reviewers and Editors,

thanks again for the reviews and comments to our article. As an attachment to this message we present the corrected version of the article.

Yours faithfully, Authors

Please also note the supplement to this comment:
https://www.clim-past-discuss.net/cp-2019-64/cp-2019-64-AC2-supplement.pdf
* * *

---

## Author Response (AR4)

**Dear Editor,**

Thank you very much for giving us an opportunity to revise and resubmit the article entitled *Droughts in the area of Poland in recent centuries in the light of multi-proxy data.* We are grateful also for all the opinions and suggestions given by the reviewers. All changes in the article were marked using red fonts. We would kindly like to inform you that the following alterations have been made to the text regarding the reviewers' comments, suggestions and opinions. For clarity the reviewers' texts are in black and our answers in blue and red.

General Remarks

**Anonymous Referee #2**

I have now reread the revised version of the articles and I consider it suitable for publication now. The authors have implemented must, albeit far from all, of the reviewer recommendations. See also my response to the authors.

The revision is much improved compared to the original submission and I can, hence, recommend publication now. I still think that a longer discussion could be make regarding whether the few droughts in in the first half of the 17th century in Poland was because of less evapotranspiration due to lower temperatures during the Little Ice Age.

The following explanation was added: Summer and winter air temperature reconstructions for Poland for the period 1401–1800 (see Przybylak 2011, 2016) indicate that thermal conditions were more favourable for the occurrence of droughts in the first half of the 17$^{th}$ century than in the period 1751–1800, which was colder. Only in the second halves of the 15th and 16th centuries were conditions better for the occurrence of summer droughts than in the first half of the 17th century. This means that the low number of droughts in the latter period is not the result of climate but is of the significantly smaller number of available sources, as we mentioned earlier.

Moreover, the Abstract could still be shortened and streamlined more. The Introduction reads considerably better now.

Answer: The text was shortened according to the Reviewer's suggestion.

Fig. 7 could still be improved graphically.

Answer: The Figure 7 was changed, and we hope that now its quality is better.

A very minor comment: I would recommend to write the "Medieval Climate Anomaly" with capitals.

Answer: Was corrected according to the Reviewer's suggestion.

**Anonymous Referee #3**

This paper written by a team of geographers, climatologists, historians and dendro-climatologists convincingly highlights the rich legacy of valuable (paleo-) climatic evidence that is available for Poland. The authors present the main features of more than 100 droughts over the last millennium including their frequency of occurrence, spatial dimension, duration and intensity. The study is based on:

• documentary data for the period 1451 to 1800, originating mostly from Pomerania and Silesia

• 22 local chronologies of tree-ring width (996 to 2015)

• 8 long-term precipitation series (1722-2015).

Droughts with the instrumental period were defined using the Palmer Drought Severity Index.

An index approach was used differentiating between extreme droughts, severe droughts and minor droughts according to duration and intensity (taking from observed hydrological and vegetative effects) at assess long term frequencies.

It is worth improving this important paper in the following respects

• A short overview over the climate of Poland is missing

Answer: A short review was added. See text below:

The climatic conditions of Poland have been characterized many times by different authors such as Paszyński and Niedźwiedź 1991; Woś 1999 and Lorenc 2005. For many years The Polish National Meteorological and Hydrological Service IMGW-PIB has presenting the fruits of their monitoring (www.klimat.pogodynka.pl), allowing analyses and assessments to be made.

The climate of Poland is in general temperate. Due to its location in the central part of the continent and being considerably affected by oceanic features in the western part of the country and a pronounced continental impact in the east, the area of Poland is diverse in terms of climatic conditions. An important geographic feature of Poland is the latitudinal course of its natural landscape types – from its sea coast in the north to its lakelands, lowlands, uplands and mountains located southward.

The mean annual air temperature in particular regions of the country varies between almost 7°C to nearly 10°C (as for the period 1981-2010) with lowest temperatures in January (from -3.5°C to 0.5°C) and highest temperatures in July (from 16.5°C to 19.5°C) (IMGW 2020). The whole country is experiencing a systematic considerable increase in air temperature with rates of increase of 0.3°C every 10 years occurring since the second half of

the 20th century. The largest increases have taken place in northern and western parts of Poland. In 2019, mean annual air temperature reached 10°C, translating into the warmest year in Poland since the beginning of instrumental measurements of air temperature. Annual precipitation ranges from 450mm in the central belt to 700 mm in the uplands and 1500-1700 mm in the highest mountain ranges in southern Poland (IMGW 2020). February is the driest month in Poland and July is the month when the highest monthly precipitation totals occur. During the last number of decades symptoms of the systematic drying of climate in Poland can be observed. Westerly and south-westerly winds predominate and only in northern, coastal parts of the country is there a considerable amount of north-westerly winds.

• The scientifically most valuable result concerns the mega-droughts, which are only partly presented and climatically not interpreted though a rich literature is known to exist.

Answer: The new information was added. See the text below and the revised version of the paper (Table 3):

The megadrought year of 1473 was detected in the Baltic Province on the basis of an oak chronology from Eastern Pomerania (Ważny, 1990). Narrow rings were observed in 80 percent of the samples for this year. The effect of the drought in 1473 can also be shifted and observed in southern Poland in 1474 (Szychowska-Krąpiec, 2010). Reconstruction based on dendrochronological data (OWDA, Cook et al. 2015) shows that, in this year, severe droughts were common in almost the entirety of Europe (but particularly in southern Germany, western Czech Republic and Austria) excluding only its northern and north-eastern parts and Spain. The drought in 1540 was observed in different parts of Europe; particularly strong evidence is available in documentary sources (Wetter et al., 2014; Pfister et al., 2015; Brázdil et al., 2016). Additionally, many dendrochronological data confirm the existence of strong droughts in much of Europe, in particular from France to Latvia, Belarus and Ukraine and from the southern Scandinavian Peninsula to northern parts of Italy (OWDA, Cook et al., 2015). Čufar et al. (2008) identified the existence of droughts in Slovenia in 1540 based on tree rings. The scale and intensity of the 1540 megadrought in Europe described by Wetter et al. (2014) as "an unprecedented 11-month-long Megadrought" (more severe than the 2003 drought in Western Europe and the 2010 drought in Russia) was, however, recently questioned by Büntgen et al. (2015), who analysed this year in light of 24,303 individual tree-ring-width measurement series. It is also worth adding here that in different parts of Europe the effect in tree rings was shifted and observed in 1541 (Büntgen et al., 2011). Analysis of our 22 dendrochronologies reveals the occurrence of narrow rings in trees growing in the Baltic Province and in the Lesser Poland Province, and thus not in the whole of Poland as shown in

the OWDA (Cook et al., 2015). In 1590, narrow rings were observed in the Baltic Province, but the decidedly strongest droughts in Europe in view of this proxy were those occurring in France and Germany (Cook et al., 2015). Narrow rings were also noted in most sites in central and eastern Europe, as well as in Scandinavia. The megadroughts occurring in Poland in the 17[th] century (1676 and 1683–84) were the least territorially extensive of all the megadroughts analysed here (see Fig. 13). Analysis of tree-ring reconstructed droughts (Cook et al. 2015) generally confirms this, except for the year 1684. In all those years strong droughts were common in Europe also, but their greatest intensity was observed in Germany, France, the Low Countries and England (1676 and 1684), but in southern Europe in 1683. The year 1748 seems to have a somewhat regional character; narrow rings were noted in the Greater Poland and Pomerania Province and in the Lesser Poland Province. There is no information about tree reaction for this drought in selected sites in central Europe (Büntgen et al., 2011). Looking at OWDA we see the occurrence of droughts in this year mainly in northern and western parts of Poland (although their severity is not so large). Evidently more severe droughts in this year in Europe were particularly observed in southern Germany, the whole of Austria and the western borders of the Czech Republic (Cook et al., 2015).

• Attempts to extrapolate the regional results to the whole of Poland are not convincing. They may be omitted.

Answer: We did not extrapolate the regional results to the whole of Poland. When information about droughts was available for all of Poland, we also count these cases for each region. Thus, the strategy is opposite to that which the reviewer suggests.

• Statistics for winter droughts should perhaps be calculated separately

Answer: After discussion we decided not to extend the article by adding further calculations, this time related only to winter droughts. However, considering the valuable opinion of the Reviewer we recognized the need to explain the conditions leading to the establishment of winter droughts in a wider context by adding the additional explanation as follows:

For comparison against the number of droughts delimited using documentary evidence, 50-year frequencies of the three categories of droughts were calculated for climatological seasons (Fig. 10). It comes as little surprise that the frequency of all-category droughts was greatest in winter. Other seasons show more-or-less similar frequencies. In winter, droughts evidently dominated in the study period in the second half of the 19th century, this is particularly well seen in the case of severe droughts, and slightly less so for moderate droughts, which were also quite frequent in the first half of the 20th century. Extreme droughts in winter do not show any significant changes over time, but it should be emphasised here that they were slightly more frequent in 1951–2000 than in 1851–1900. Moreover, in addition to winter droughts it should be pointed out the deficit in precipitation during this season is usually connected to temporarily increasing continentality of climate conditions which are related to

the advection of very cold and dry polar continental air masses from the east, sometimes even with the mixture of very cold arctic air masses. During such conditions deep soil frost increases which does not allow the water infiltration into deeper layers. Thus, almost all melting snow is transformed into spring surface run-off volume and only the negligible part of this volume is transformed into groundwater. Such conditions may lead to the occurrence of very dry spell in spring. In spring, moderate droughts prevailed still in the period 1851–1950 (usually 4–6 cases), with a greater frequency in the first 50-year period. Both severe and extreme droughts were most frequent (usually 1–3 cases) in 1851–1900, and in particular in 1951–2000 (Fig. 10). In summer there is a clear change in the time pattern of drought occurrence: drought frequency rises in the 20th century (except severe droughts), and in the case of moderate droughts particularly in its second half. Frequency of extreme droughts is evidently higher in the 20th century compared to pre-1900 period. In autumn, moderate droughts do not show great changes in the last two centuries, while severe and extreme droughts were most frequent in the 20th century (Fig. 10).

Fig. 11 is dominated by droughts with a duration of 2 months. Which is the criterion for "extreme" in this case?

Answer: The information is given on page 22, lines 9–13, see in particular the last sentence in red:

„Secondly, a drought was considered to be at least two consecutive months during which the SPI1 value was ≤-0.50. Thus identified, a drought was determined both in terms of duration and by category. Thirdly, drought category was determined by the dry month of lowest SPI1 value. A drought was thus considered extreme if the SPI1 value for at least one of the drought months was ≤-2.00."